# Distinct solvation patterns of OH⁻ versus H₃O⁺ charge defects at electrified gold/water interfaces govern their properties

Chanbum Park [1] ✉, Soumya Ghosh[1,3], Harald Forbert[2] & Dominik Marx[1]

Understanding the solvation structures of OH⁻ and H₃O⁺ at metal interfaces is crucial for developing efficient electrochemical devices. In this paper, we present a detailed investigation of the solvation structures of OH⁻ and H₃O⁺ near gold electrodes under alkaline and acidic aqueous conditions, using ab initio molecular dynamics simulations at controlled surface charge density conditions. Our findings reveal that the adsorption tendencies of OH⁻ and H₃O⁺ are strongly influenced by the oscillating net atomic charge of water normal to the electrified interface in concert with the distinct solvation patterns of these charge defects. While OH⁻ preferentially adsorbs onto the gold surface within the first water layer, the positive net atomic charge restricts the closest approach of H₃O⁺ to beyond the first water layer. We unveil resting and active states that support charge transfer processes at the gold/water interface, which critically involve Au atoms in a unique Grotthuss-like mechanism.

Electric double layers, formed at the interface between a solid electrode and a liquid electrolyte, are of paramount importance for determining performance and lifetime of electrochemical devices[1–3]. In particular, metal/water interfaces are gaining attention for their use in electrochemical energy conversion and storage devices[4–7]. Although the hydrogen evolution reaction (HER), hydrogen oxidation reaction (HOR), and oxygen evolution reaction (OER) are fundamental to the efficiency of electrochemical devices[8–10], there are still long-standing fundamental questions to be resolved. This gap in knowledge can be attributed to the lack of molecular understanding of electric double layers despite decades of intense research. The electric double layer is buried between electrodes and electrolytes, making it challenging to experimentally probe aqueous electrolytes at metal interfaces. Experiments and computational studies have successfully provided a wealth of insights into the double-layer structure, yet most studies have focused on the structural properties of water molecules[3,11–13].

Notably, understanding the solvation structures of the water ions in alkaline or acidic aqueous electrolytes at metal interfaces, namely H₃O⁺ and OH⁻, is crucial for comprehending the pH-dependent

reaction mechanisms and kinetics in HER, HOR, and OER. Previous computational studies using ab initio molecular dynamics (AIMD)[14] simulations found that the hypercoordinated solvation structure of the "proton hole", OH⁻, subject to four accepted hydrogen bonds (HBs) as characteristic in bulk solutions[15–17], survives at non-metallic aqueous interfaces[18–20]. In this configuration, the oxygen of OH⁻ is at the center of a square formed by four proton-donating water molecules within slit pores offered by layered minerals, which is overall very similar to bulk solutions while featuring a dangling OH state in the interfacial case. Yet, it has been discovered that this peculiar solvation structure significantly impacts ion transport in such slit pores[18]. Recent AIMD simulation studies found similar hypercoordinated solvation patterns of OH⁻ within aqueous interfaces at graphene, boron nitride, and NaCl surfaces[19,20].

These previous findings raise a series of important questions: Does OH⁻ also hold a similar hypercoordinated HB structure at metal/water interfaces? If so, does that solvation pattern change once external bias is applied as relevant in electrochemistry? How does the specific solvation motif impact charge migration at such electrified interfaces? Last but not least, is the solvation structure of the cationic

[1]Lehrstuhl für Theoretische Chemie, Ruhr-Universität Bochum, 44780 Bochum, Germany. [2]Center for Solvation Science ZEMOS, Ruhr-Universität Bochum, 44780 Bochum, Germany. [3]Present address: Tata Institute of Fundamental Research Hyderabad, Hyderabad 500046 Telangana, India. ✉e-mail: chanbum.park@theochem.ruhr-uni-bochum.de

counterpart, the hydrated excess proton $H_3O^+$, different within metal/water interfaces compared to bulk solutions?

When it comes to the hydrated proton, elucidating the solvation structure of $H_3O^+$ at metal/water interfaces as a function of bias is crucial to understand the HER mechanisms in electrochemical devices. An early investigation suggested that the rate-limiting step for proton transfer to the metal surface is caused by the re-orientation of the water molecule in the first layer, which accepts a proton from an $H_3O^+$ ion in the second layer[21]. This mechanism has been believed to underlie the Volmer reaction in HER, and a similar mechanism was observed at silver or platinum electrodes using an empirical valence bond model[22,23]. However, a recent AIMD study finds that the source of the adsorbed hydrogen atom at the metal surface is the dissociation of water molecules in the first layer rather than the $H_3O^+$ ion, and that the HB connectivity is crucial for determining the reaction kinetics[24]. Although there are studies that elucidate the behavior of $H_3O^+$ (see refs. 20,24–28) and $OH^-$ (see ref. 19) ions at metal surfaces, their intrinsic solvation structures remain unexplored at the atomistic level.

Likewise, comprehensive atomistic-level insights into the electrolyte structure at electrified metal/water interfaces are still lacking at finite bias conditions, as relevant in electrochemical environments. To simulate an electrochemical cell subject to bias potentials in the framework of AIMD simulations[14], various methods have been developed, including charged slab, effective screening medium, constant Fermi-level, finite field, charge imbalan, or grand canonical methods[26,29–33]. Recently, a method using an inert computational neon counter electrode has been introduced[34]. Here, we employ this technique to carry out AIMD simulations of "pristine" $H_3O^+$ (hydronium) and $OH^-$ (hydroxide) charge defects in pure water at electrified interfaces, where the surface charge density can be continuously controlled computationally. Crucially, no additional electrolyte ions are present in the simulation supercell. This particular approach ensures that the water structure at the metal surface—particularly its interfacial HB network—does not get perturbed by the presence of electrolyte ions. Such ions at aqueous surfaces are known to strongly alter the interfacial water structure due to the formation of their own hydration shells, which can interfere with those of the $H_3O^+$ and $OH^-$ charge defects, depending on the ion concentrations near the electrode. The drawback of this setup is the absence of an electric double layer and the associated double-layer capacitance at the gold/water interface. Because of the lack of screening counterions in the electric double layer, the computational voltage does not represent the local voltage drop near the metal electrode. However, these features would be essential for modeling realistic electrochemical cells, where a well-defined relationship between electrode potential and surface charge density is required. Hence, the underlying limitations must be taken into account when using a similar computational setup in future electrochemical studies. In contrast to such applications, our aim here is to set up an electrified interface between an explicit gold electrode model in contact with pure water, enabling us to investigate—depending on surface charge density—the intrinsic structural and dynamical properties of $H_3O^+$ and $OH^-$ charge defects. This notably includes the specific charge transfer reactions of these defects within interfacial water close to negatively and positively charged metal electrodes, which underlie their distinct "pristine" Grotthuss-style migration mechanisms in such idealized acidic and alkaline aqueous environments, respectively.

In this work, we implemented the method[34] in the CP2K simulation package[35–37] to investigate Au(111)/water interfaces in acidic and alkaline solutions at different surface charge density conditions to unveil the solvation structures of $OH^-$ and $H_3O^+$ at these electrified interfaces in the absence of any perturbing electrolyte ions solvated within the interfacial HB network; note that, for simplicity, we report our findings in terms of the formal computational voltages rather than the corresponding surface charge densities, which are provided in Supplementary Table 1. Our findings reveal that the net atomic charge (NAC) of water molecules near the Au interface governs the adsorption behavior of the $H_3O^+$ and $OH^-$ charge defects. Polarization and charge transfer effects between the metal surface and water molecules at the interface induce a positive net charge at the first layer, creating attractive forces for $OH^-$. Conversely, the positive net charge seems to restrict the closest approach of $H_3O^+$ to one solvent layer beyond the interface. Importantly, we discovered previously unknown adsorption and solvation complexes of $OH^-$ at the Au interface, where the $OH^-$ defect structure commensurably anchors on top of two neighboring Au atoms in the active state for charge transfer. This not only leads to a unique Grotthuss-like mechanism that critically involves atoms from the electrode, but also significantly decreases the charge transfer rate involving $OH^-$ defects compared to bulk. Since the solvation states and charge transfer scenario for $H_3O^+$ are found to remain essentially unaffected, we disclose here distinct differences for charge transfer processes in alkaline versus acidic gold/water interfaces with reference to their bulk environments.

## Results

The diffusion of $H_3O^+$ and $OH^-$ ions towards an electrified interface is fundamental to many electrochemical reactions. Understanding how ions approach the electrode is critical for elucidating these processes. In this study, we examine the electronic and structural properties of $H_3O^+$ and $OH^-$ ions in aqueous solution at the electrified Au/water interface; representative snapshots of the solvated hydronium and hydroxide ions are provided in Fig. 1a and b, respectively. To simulate our electrified interfaces, we adapt a method with an explicit computational neon counter electrode, which enables AIMD simulations under finite computational bias potentials, allowing control over the surface charge density of the metal electrode[34]; details of this methodology and the charging procedure are provided in Supplementary Note 1 and Supplementary Figs. 1 and 2 in the Supplementary Information (SI). Specifically, we apply this AIMD technique to the large $HClO_4$ and NaOH systems depicted in Fig. 1a and b to unravel the intrinsic properties of the pristine $H_3O^+$ and $OH^-$ charge defects at gold electrodes in contact with pure water, as a function of the surface charge density of the electrode. Having this clear focus in mind, we stress that our maximum (positive and negative) surface charge densities (see below) are chosen such that we safely stay away from any interfering electrochemical reactions in both cases. Furthermore, we deliberately do not include any electrolyte ions to avoid perturbations of the HB network of the interfacial water layers as a result of hosting their solvation shells.

### Properties of the hydrated proton at the interface

To unveil the migration process of $H_3O^+$ toward the electrified Au electrode subject to a negative surface charge density, we track the positions of $H_3O^+$ as depicted in Fig. 1c using a representative trajectory (see Supplementary Fig. 3 for additional data); the relation of the computational bias potentials and the generated surface charge densities is outlined in Supplementary Note 1 and visualized in Supplementary Fig. 2. At the initial stages of migration, $H_3O^+$ appears to be unaffected by the applied bias. After some time, it starts directional diffusion towards the Au electrode via the well-known Grotthuss mechanism[16,38,39]. As a result of this charge migration mechanism, the atomic constituents of $H_3O^+$ (and similarly of $OH^-$, see below) vividly change during AIMD simulations. Therefore, tagging and thus tracking of the charge defects (rather of their constituting atoms) has been realized using a continuous collective variable which includes all O and H atoms in the system (except for $ClO_4^-$) as explained in Supplementary Note 2. The $H_3O^+$ ion reaches the interface (IF) region (see caption of Fig. 1 for definition) at some stage (here around 23 ps), but shortly thereafter it moves back to the second water layer where it prefers to reside. In the limit of zero surface charge density, $H_3O^+$ exhibits non-

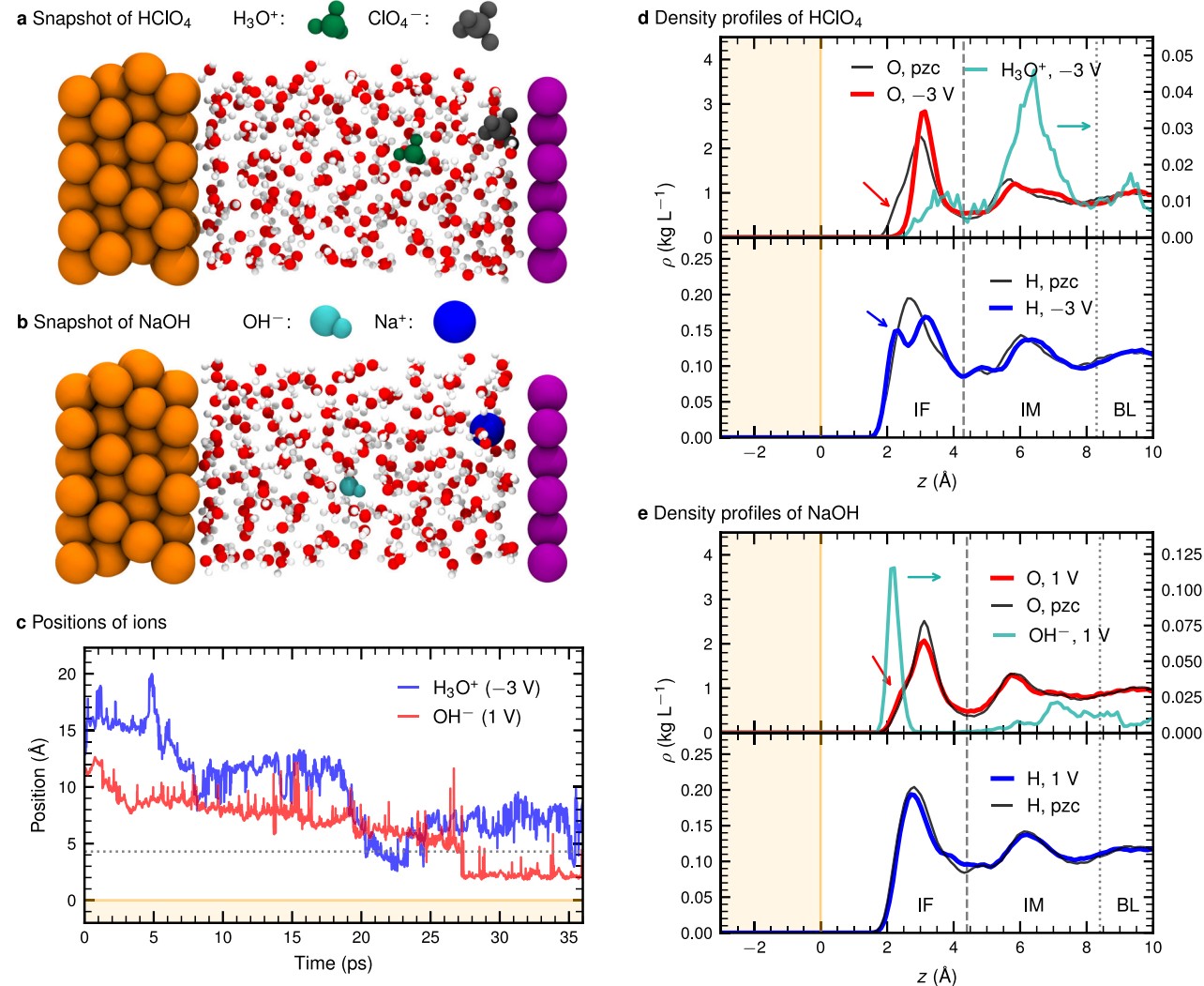

**Fig. 1 | Simulation cell setups, charge migration tracking of the H₃O⁺ and OH⁻ charge defects and density profiles for acidic and alkaline aqueous electrolytes at Au interfaces at finite and zero surface charge density conditions corresponding to finite bias potential and pzc conditions as indicated.** The surface charge densities corresponding to the reported bias potentials are compiled in Supplementary Table 1. Configuration snapshots of the simulation cells hosting dissociated **a** HClO₄ and **b** NaOH in water between the gold electrode (left) and the inert neon-based computational counter electrode (right)[34]. **c** Representative migration pathways of the H₃O⁺ and OH⁻ charge defects (see text for the tagging approach) from the bulk-like regime (visualized by the snapshots in panel a and b) to the electrified interfacial region as a result of applying a potential bias of −3 and +1 V, respectively; see Supplementary Note 1 for definition of the computational bias potentials and their relation to the surface charge densities. The horizontal dashed line delimits the interfacial water layer according to the density profiles in panels d and e. Density profiles of all O, H, and Au atoms as well as of the O sites of H₃O⁺ and OH⁻ along the surface normal z referenced to the top-most Au layer for the **d** acidic and **e** alkaline solutions visualized in panels a and b at −3 and +1 V, respectively; the potential of zero charge (pzc) results are shown for reference, and arrows highlight features discussed in the text. The interfacial (IF) and bulk-like (BL) regions are indicated by vertical dashed and dotted lines, respectively, while the intermediate (IM) region is in between as indicated by the respective labels in panels d and e.

directional diffusion between the two electrodes (see Supplementary Fig. 4). Overall, we consistently find that H₃O⁺ does not prefer to integrate itself into the first water layer at the gold electrode, regardless of whether the surface is charged or uncharged.

How does the H₃O⁺ charge defect integrate itself into the layered water structure of gold/water interfaces? In Fig. 1d, the density distribution of the oxygen and hydrogen sites of the water molecules subject to negative surface charge density and at pzc conditions is compared with the density profile of H₃O⁺ along the surface normal z referenced to the top-most Au layer. At the negatively charged Au surface, the O density profile shifts away from the surface and concurrently sharpens compared to the pzc reference (indicated by a red arrow in the upper panel of Fig. 1d). The shifted O peak is due to interactions with hydrogen atoms of the water molecules that are pointing towards the Au surface, indicated by a blue arrow in the lower

panel of Fig. 1d. Thus, the H peak is found to split at negative bias potential: The peak closer to the surface corresponds to those H atoms of water molecules that are pointing directly towards the metal surface, while the peak further out represents H atoms that interact with other water molecules via HBs. Overall, the first solvation layer of water molecules features distinct structural changes in response to the negatively charged gold electrode. Given that interfacial water scenario at negative surface charge conditions, we find that H₃O⁺ resides preferentially outside the IF region, i.e., beyond the dashed vertical line. Yet there is some residual contribution found within the IF regime mainly located in-between the first and second water layers as provided by the respective O peaks (see Fig. 1d).

What is the orientation of H₃O⁺ if it is located in the IF region? The answer is given by analyzing the angle distributions in Fig. 2a and b separately for the Eigen and Zundel states of the hydrated proton. It is

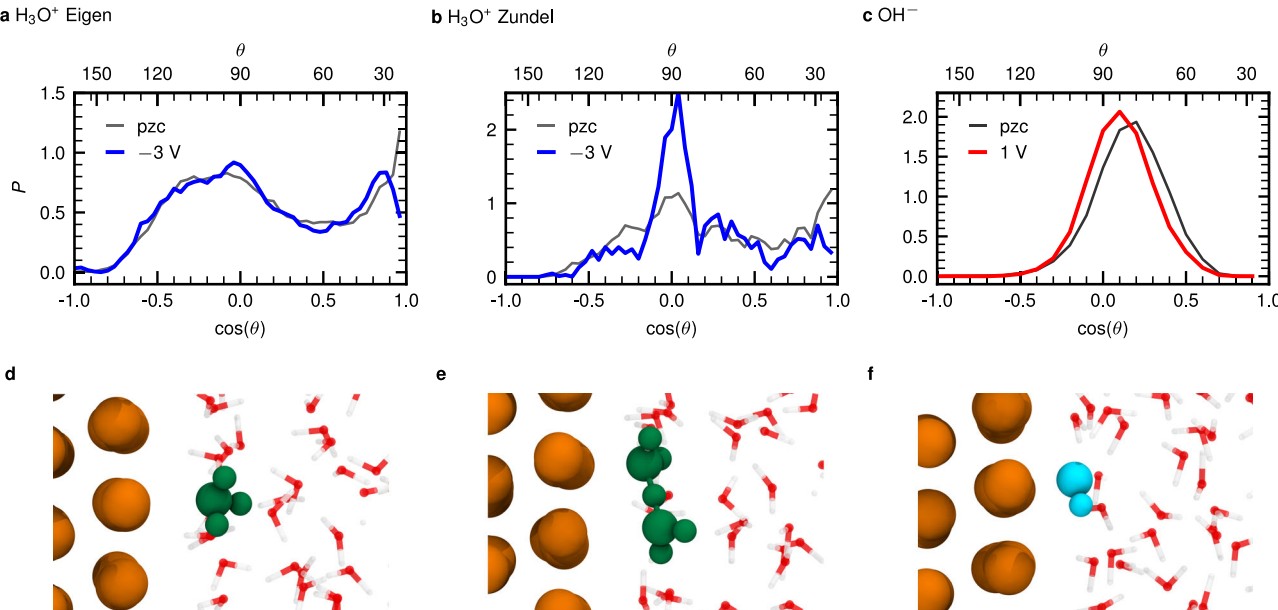

**Fig. 2 | Orientations of H₃O⁺ (split in Eigen and Zundel states, see text) and OH⁻ defects in the interfacial region at finite and zero surface charge density conditions corresponding to finite bias potential and pzc conditions as indicated; the surface charge densities corresponding to the reported bias potentials are compiled in Supplementary Table 1.** Probability distribution functions of the cosine of the angle $\theta$ of **a** the OH vectors in H₃O⁺, **b** the angle formed by two oxygen atoms in a Zundel state, and **c** the OH vector of OH⁻ all with respect to the surface normal without (pzc) and subject to the indicated bias. Representative snapshots visualizing the most probable arrangements are shown in the lower panels **d**, **e**, and **f**.

well known that the structural diffusion or Grotthuss mechanism of H₃O⁺ involves a three-fold HB hydronium complex, dubbed Eigen state, and a proton-shared dimer complex, the Zundel state[16,38,39]. Our analysis indicates that the structure of H₃O⁺ within the IF region is not significantly influenced by the surface charge density in the investigated regime regardless of whether it is in the Eigen or Zundel state. However, we note the suppressed probability of $\theta$ angles at less than about 30° in the Eigen state and the enhancement of $\theta \approx 90°$ orientations in Zundel-like complexes compared to their pzc references. Figure 2a unveils that the three OH vectors in the Eigen complex of H₃O⁺ are either close to parallel to the surface, as illustrated in Fig. 2d or pointing to the bulk phase with the oxygen atom of H₃O⁺ facing the surface (see Supplementary Fig. 7). Interestingly, this orientation closely resembles the H₃O⁺ configuration observed experimentally[40,41].

Simultaneously, the OH vectors of the IF water molecules predominantly point towards the Au surface at finite bias, resulting in an increased number of dangling OH bonds compared to the pzc reference. This effect becomes more pronounced as the surface charge density increases, reflecting qualitatively the response of water orientations to finite external bias (see Supplementary Fig. 5a). Moreover, a detailed decomposition of the HB patterns of the individual water molecules in the IF layer unveils that the HB states also depend on the surface charge, whereas they are identical in the BL regime (see caption of Fig. 1 for definition), which features a distinctly different overall pattern dominated by molecules that accept and donate two HBs (D2A2 species), see Supplementary Fig. 6. On the other hand, H₃O⁺ in its Eigen state prefers to form HBs with surrounding water molecules rather than having a dangling OH pointing to the Au surface, in stark contrast to the water molecules. We find that this is very similar at zero surface charge density. Thus, the Eigen state in interfacial water remains virtually unperturbed compared to bulk water.

The arrangement of the Zundel complex within the IF region is extracted from Fig. 2b where the angle is obtained from the vector formed by two oxygen atoms in the Zundel state with respect to the surface normal. The angle distribution discloses that the Zundel complex is likely to align parallel to the surface, which is not affected

by the surface charge. As presented in snapshot Fig. 2e, the flat Zundel complex is embedded within the stratified HB network, much like in bulk water.

Overall, the structure of the hydrated excess proton in the IF region is found to be rather insensitive to the surface charge density in the investigated regime. Both the Eigen- and Zundel-like solvation structures of H₃O⁺ are found to be well-integrated in the HB network of the IF water layer, which imparts orientational stability toward field-induced reorientation at the given surface charge density. Similar conclusions will be reached for the hydrated proton hole defect as to the orientation of the OH vector with respect to the surface normal, see below. This general picture is expected to change when charging the metal surface even more by increasing the computational bias potential (thus inducing electrochemical reactions) or due to the presence of additional electrolyte ions in the first solvation layer of the metal electrode (thus breaking the HB network herein).

Additionally, the radial distribution functions (RDFs) and free energy profiles for charge transfer in Fig. 3a and d show that the properties of the excess proton within the IF region at the charged gold/water interface remain essentially unaltered with respect to those at the uncharged electrode (corresponding to the pzc condition) as well as to those in the bulk-like region (BL as defined in Fig. 1); see Supplementary Fig. 8 for the comprehensive set of RDFs split into Eigen and Zundel states as well as IF and BL regions at negative surface charge densities versus pzc conditions.

**Properties of the OH⁻ charge defect at the interface**

While the excess proton defect, H₃O⁺, tends to remain within the IM and BL regions (see caption of Fig. 1 for definitions) rather than populating the IF regime even at finite bias (see Fig. 1 for the definitions of these regions) the behavior of the proton hole defect, OH⁻, is found to be significantly different. First of all, Fig. 1c reveals that OH⁻ in the presence of the positively charged electrode dives into the first water layer and remains localized therein (see Supplementary Fig. 9 for additional analysis). Once OH⁻ enters the first water layer, it shows a strong tendency to sit, on average, much closer to the Au surface than

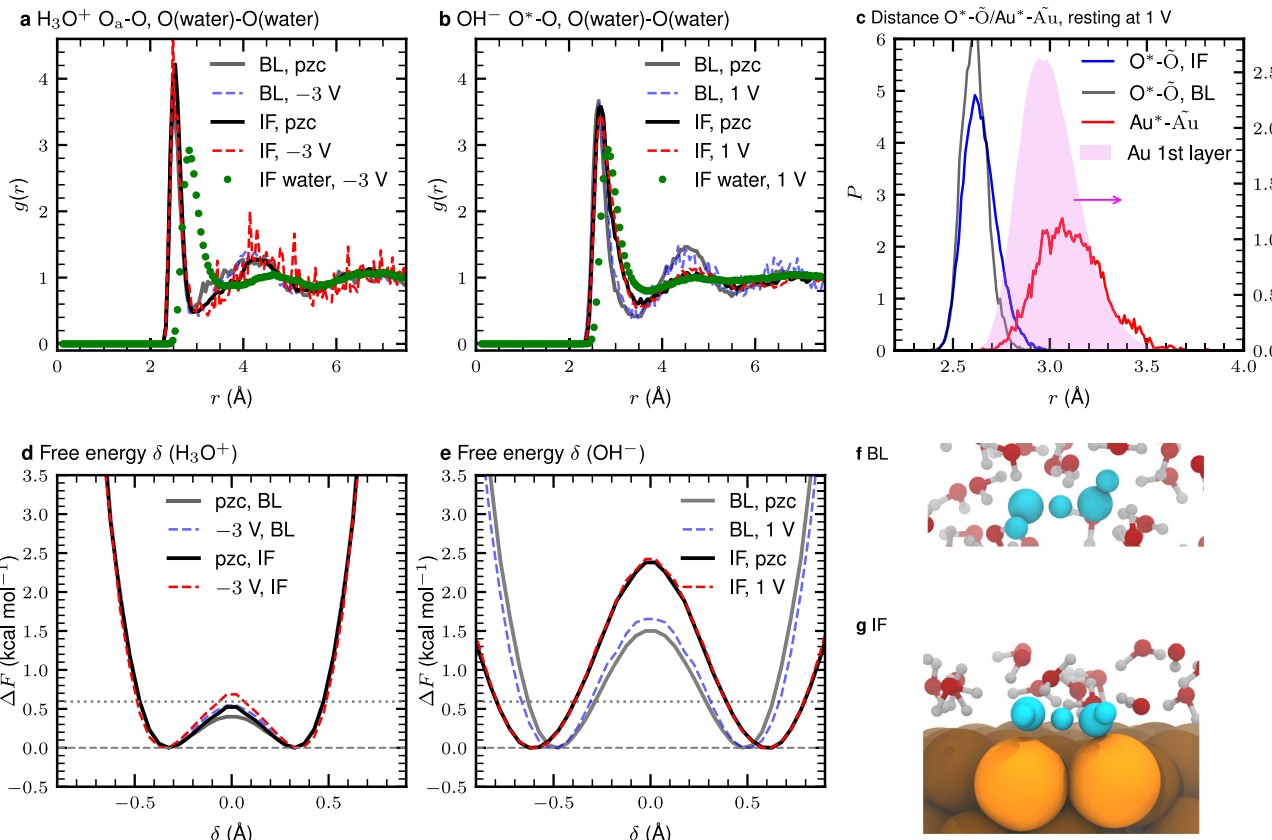

**Fig. 3 | Radial distribution functions, charge transfer free energies of H₃O⁺ and OH⁻ and structural analysis at finite and zero surface charge density conditions corresponding to finite bias potential and pzc conditions as indicated.** The surface charge densities corresponding to the reported bias potentials are compiled in Supplementary Table 1. RDFs of **a** $O_a$ (H₃O⁺) and **b** O* (OH⁻) with respect to the oxygen (O) atoms of the water molecules in the interface (IF) and bulk-like (BL) regions, respectively; here $O_a$ and O* indicate the oxygen atom in the H₃O⁺ and OH⁻ ions, respectively. The corresponding IF water-water O⋯O RDFs are shown for reference at the respective finite surface charge density conditions; the pzc data are very similar in both cases and thus not shown. **c** Distribution function of the distance between O* and Õ in the IF and BL regions as well as Au*⋯Ãu and Au⋯Au distances in the top Au layer, where Au* and Ãu denotes the closest Au atom to O* and Õ, respectively, as defined in the text. The average O*⋯Õ (IF), O*⋯Õ (BL), Au*⋯Ãu and Au⋯Au (1st layer) distances are 2.66, 2.63, 3.14, and 3.02 Å. Free energy profiles along the charge transfer coordinate $\delta$ for **d** H₃O⁺ and **e** OH⁻ (see the definition of $\delta$ in the main text). The horizontal dotted lines indicate the thermal energy $k_B T$. Representative snapshots of proton hole transfer events resulting in charge migration of OH⁻ within the **f** BL and **g** IF regions.

the water molecules (Fig. 1e and Supplementary Fig. 12). A similar behavior has been observed at the more reactive platinum surface in the absence of any potential bias, where OH⁻ stemming from self-ionization of water adsorbs at the Pt surface as OH, but not for gold[28]. Clearly, the positive surface charge density drives the OH⁻ defect− initially located deep in the BL regime (at approximately 10 Å from the topmost Au layer at time zero in the migration pathway according to Fig. 1c)−into the IF region, which extends up to about $z \approx 4.4$ Å (as indicated by the water density profile in Fig. 1e). The defect crosses this distance at $t \approx 27$ ps (Fig. 1c) before finally localizing just $z \approx 2$ Å from the surface. We find that the interfacial water structure itself, as quantified by the O and H density profiles in Fig. 1e is not impacted by the positive surface charge density compared to the uncharged electrode representing pzc conditions (see Supplementary Fig. 11 for refined analysis). This OH⁻ migration scenario is distinctly different from H₃O⁺ where this defect barely moves into the IF region even at the negatively charged gold electrode. Rather, it ends up in the IM regime or at best in between the first and second solvation layer as observed for $t > 20$ ps in the representative pathway of Fig. 1c.

Given the significant difference of OH⁻ with respect to H₃O⁺, it is intriguing to investigate the structure of OH⁻ at the Au surface. Is the orientation of OH⁻ at the metallic Au surface similar to the perpendicular one known from layered minerals and the graphene/water interface? The probability distribution of the OH vector orientation at

the Au surface in Fig. 2c shows that OH⁻ is roughly aligned with an O−H angle of approximately 80° with respect to the surface independently from the surface charge density (a representative snapshot is shown in Fig. 2f and refined analyses of mass density profiles as well as the OH vector orientation is provided in Supplementary Fig. 13 for different surface charge densities compared to the pzc reference. Thus, this orientation at the gold/water interface strongly contrasts with what is known for OH⁻ close to mineral and graphene interfaces[18,19]. At this point, we anticipate a key difference of OH⁻ in contact with the gold surface: We show below that OH⁻ strongly interacts with individual Au atoms (namely Au* and Ãu, as analyzed below, through one of the lone pairs of its oxygen atom) in concert with establishing significant H-bonding to several surrounding water molecules to stabilize itself in the preferred hypercoordinated state while including gold atoms. Additionally, comparison of the RDFs of OH⁻ in the IF and BL regions, see Fig. 3b, reveals that the solvation structure of OH⁻ in the first water layer (IF region) is distinctly different from that in the bulk liquid (BL regime).

It is known that water molecules can form two-dimensional HB networks at planar surfaces[11,32,42]. This finding might suggest that such networks enhance structural diffusion processes compared to the bulk environment. In particular, one might speculate that the orientation of OH⁻, which is the same direction as the HBs in such two-dimensional networks of water molecules, facilitates proton hole diffusion at the Au

surface. However, Fig. 3e discloses the opposite picture: The free energy barriers for proton hole transfer in the IF region increase significantly as compared to those in the BL region, both at zero and finite surface charge density conditions. Moreover, the free energy minima are significantly further apart in the first water layer (IF region) as compared to those in bulk (BL region), implying that the proton hole will have to traverse a longer distance along with a much higher activation free energy; we note here that a recent study in the absence of bias has shown that quantum delocalization of protons is negligible at the gold/water interface while relevant in the case of platinum[28]. Overall, we find that the free energy barrier to charge transfer via hydroxide defects at the gold/water interface (IF data in Fig. 3e) increases by 50–60% compared to the bulk reference (BL data therein). This corresponds to a decrease of the charge transfer rate that approaches an order of magnitude effect, namely to about 0.2–0.3 that of the bulk rate according to a simple transition state estimate.

The increased distance between the minima along the charge transfer coordinate in the IF versus BL regime (see Fig. 3e) suggests analysis of the probability distribution of the distance between O* (of O*H⁻) and the hydrogen-donating oxygen $\tilde{O}$ (for proton transfer from the respective water molecule toward O*H⁻, thus forming H$_2$O* after transfer) separately in the IF and BL regions in Fig. 3c. Comparison reveals that the average distance between O* and $\tilde{O}$ elongates when OH⁻ is localized in the IF region. Besides, the distance between Au* (the nearest Au atom to O*) and $\tilde{Au}$ (nearest Au to $\tilde{O}$) is also significantly increased compared to the other Au–Au distances within the first Au layer (see also Fig. 3c).

These important findings indicate that OH⁻ and the oxygen of the very hydrogen-donating water molecule involved in the charge transfer event are strongly correlated with the two nearby Au atoms as shown by the snapshot in Fig. 3g. This picture illustrates that when the charge transfer event occurs, the two oxygen atoms are bridged by the active H* while they remain anchored at the two Au atoms. In other words: These two Au atoms serve as the pillars of the (HB-) bridge along which the proton transfer occurs, thus resulting in charge transfer of the OH⁻ defect along one HB. The long Au*–$\tilde{Au}$ distance stretches the HB-bridge O* ⋯ H* ⋯ $\tilde{O}$ in the anchored complex which translates into the longer separation between the minima in the free energy profile and, crucially, generates the increased activation free energy for charge transfer when OH⁻ is in the IF region (see Fig. 3e). Consequently, these processes in the IF regime enormously hinder charge transfer involving OH⁻ in the first water layer on the gold surface.

It is by now evident that H$_3$O$^+$ and OH⁻ ions at electrified gold/water interfaces feature vastly distinctive behaviors: While the positive charge defect remains essentially unperturbed concerning structure and activation free energy for charge transfer compared to the situation in bulk, most notably the charge transfer barrier for OH⁻ is strongly increased at the interface. This prompts us to investigate the electronic properties at the Au interface with a particular focus on possible correlations with the strongly perturbed solvation pattern of OH⁻ at the electrified interface.

## Electronic properties and solvation structures

Analyzing the density-derived electrostatic and chemical (DDEC) charges[43] provides deep insights into electronic properties of the charge defects at gold/water interfaces. Figure 4a exhibits the excess charges of oxygen and hydrogen atoms of all water molecules as well as the total net charge of the OH⁻ and H$_3$O$^+$ ions along the surface normal with reference to the respective density profiles at positive and negative surface charge densities, respectively.

The excess charge of the solvent features a damped oscillatory decay toward zero from the IF to the BL region in both cases. Solely due to the interaction of the water molecules with the Au electrode, there is a significant positive excess charge of roughly 0.3 $e$ very close

to the metal surface which quickly drops to a negative value of about − 0.2 $e$ that rises again to ≈ 0.1 $e$ all within the IF region (being the first solvation layer from about 1.3 to 4.3 Å as marked), before Δ$q(z)$ decays to zero (in an oscillatory manner) far from the electrode.

Importantly, we observe the very same behavior for neat water at zero surface charge density (pzc) conditions in Fig. 4a, thus the phenomenon is not induced by localizing charged species in the IF region due to charging the electrode. Meanwhile, the fractional values of the total net charge of H$_3$O$^+$ and OH⁻ localized in the IF regime are roughly 0.5–0.6 and − (0.3–0.5) $e$, respectively, which is consistent with a previous study of H$_3$O$^+$ using three solvent layers of eight molecules each[44].

Concerning the preferred location of the two charge defects at the electrified gold/water interface, the full picture is obtained only after considering at the same time the excess charge profile Δ$q(z)$, the total net charges of the H$_3$O$^+$ and OH⁻ ions along $z$ as well as their density profiles (all shown in Fig. 4a) together with the oxygen and hydrogen density profiles shown in Fig. 1d and e. Given these profiles, the OH⁻ anion with its negative charge of about − (0.3-0.5) $e$ prefers to localize itself right in the region of pronounced positive excess charge, which is extremely close to the gold electrode within the first solvation layer. In contrast, the H$_3$O$^+$ cation, carrying a positive charge of 0.5-0.6 $e$, resides at much larger distances, close to the first minimum of the water density, where the excess charge is negative or small (see Supplementary Figs. 10 and 11 for further details). Furthermore, the charge transfer between the H and O atoms of water and Au is found to depend on the number of HB that are donated by the IF water molecules (see Supplementary Figs. 14 and 15). Overall, this analysis explains why H$_3$O$^+$ tends to avoid the first solvation layer; in stark contrast, the OH⁻ charge defect, prefers to reside extremely close to the gold electrode.

Since OH⁻ shows a very distinct arrangement at the Au surface, we further investigate the average NACs of OH⁻, Au* and of all Au atoms in the second-nearest neighbor environment of Au* in Fig. 4b. We find that the net charge of OH⁻ is fairly constant in the different solvation structures visualized in Fig. 4f–i, except in D$_1$A$_4$ as a result of accepting four HBs resulting in increased charge transfer toward OH⁻. Importantly, the NAC of Au* is positive, hence, our finding is that Au* can be considered as a donating atom that solvates OH⁻ much like a proton of a water molecule that donates a HB to OH⁻. We therefore discover that surface gold atoms interacting with the oxygen site of OH⁻ take over the role of water molecules that stabilize the proton hole defect in aqueous environments.

To support this idea, we now analyze in detail the HB statistics of OH⁻ within the IF region in Fig. 4c. It is well-known that OH⁻ in bulk water is solvated by about 4 to 5 water molecules[15–17]. In this "resting state", OH⁻ accepts on average a bit more than four HBs at its O end and donates transiently one HB via its hydroxyl hydrogen, resulting in the hypercoordinated state of hydroxide in water[15–17]. In the "active state" in which charge transfer occurs, the number of accepted HBs is reduced to a bit more than three, while the donated HB is strengthened[15–17]. The two states can be distinguished based on the charge transfer coordinate $\delta = d_{O^*-H^*} - d_{\tilde{O}-H^*}$, where H* is the transferred proton which moves along the respective HB from an intact water molecule with $\tilde{O}$ oxygen to the O* oxygen of OH⁻, using $|\delta| > 0.5$ and $|\delta| < 0.1$ to define the resting and active states of OH⁻, respectively[15–17]. We confirm this well-established picture in the BL region of our setup independently from the surface charge (numbers are reported in Supplementary Table 2). In stark contrast, in the IF region when considering the donor (D) or acceptor (A) number in the resting state of OH⁻ with respect to its solvating water molecules, see Fig. 4c, we find that on average about one HB is donated but only roughly three are accepted by OH⁻, thus about one HB is missing compared to the preferred hypercoordinated state of OH⁻. But interestingly, the nearest neighbor gold atom, Au*, which we know is

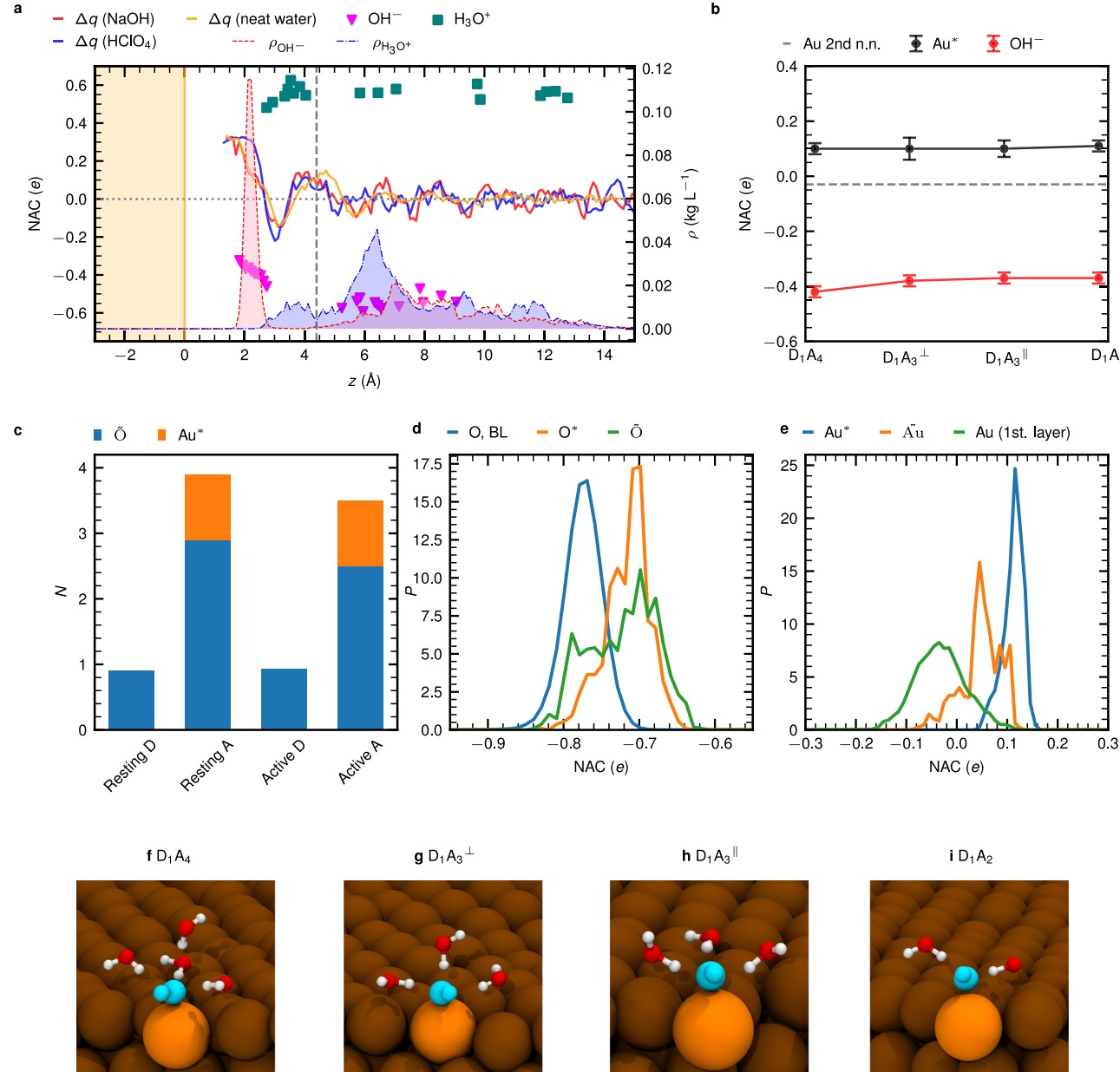

**Fig. 4 | Net atomic charge analyses and hydrogen bond statistics of the OH⁻ and H₃O⁺ charge defects in the NaOH and HClO₄ aqueous solutions close to the gold/water interface at finite and zero surface charge density conditions corresponding to a computational potential bias of +1 and −3 V, respectively; See Supplementary Table 1. a** Excess charge profile $\Delta q(z)$ computed from the net atomic charge (NAC) of the oxygen and hydrogen atoms of the $H_2O$ molecules along the surface normal in the alkaline (solid red line) and acidic (solid blue line) electrolytes systems and for reference in neat water at zero bias (solid orange line). Downward triangles and squares represent the total net charge of the OH⁻ and H₃O⁺ ions, respectively. The density profiles $\rho(z)$ of OH⁻ and H₃O⁺ (right $y$-scale) are shown in dashed red and dash-dotted blue lines, respectively, including shading of the corresponding areas to enhance the visibility. **b** Average NAC of Au* and of the second-nearest neighbors (2nd n.n.) of Au* as well as average total net charge of OH⁻ obtained from the NACs of its H and O atom depending on the hydrogen bonding state of OH⁻ being in the IF region; $D_nA_m$ denotes that OH⁻ donates $n$ and accepts $m$ HBs while ⊥ and ∥ define the two distinct orientations of the $D_1A_3$ state according to panels g and h, respectively. Error bars indicate the standard deviation obtained by analyzing ten independent frames. **c** Average number $N$ of donor (D) and acceptor (A) HBs of OH⁻ in its "resting" and "active" state (blue bars); see text. Orange bars on the top of the blue bars indicate the number of nearby Au atoms (see text). **d** Distribution function of the NAC of O*, Õ and O (in the BL region) in the resting state of OH⁻. **e** Distribution function of the NAC of Au*, Ãu and Au in the first layer while OH⁻ is in the resting state. **f–i** Representative $D_nA_m$ solvation states of OH⁻ near the Au surface analyzed in **b**; Au* is highlighted in bright orange while the water molecules which are not associated with the solvation shell of OH⁻ (cyan) are not presented.

positively charged according to Fig. 4b, is found to fully compensate for that missing HB, see orange contribution in Fig. 4c (analysis at zero surface charge density is presented in Supplementary Table 2).

Therefore, we clearly unveil that Au* acts like a hydrogen-donating species that stabilizes OH⁻ at the Au surface together with the solvating water molecules. Only this mechanism generates the

preferred hypercoordinated state of OH⁻ also at the electrified gold/water interface as visualized by the configurations in Fig. 4f–i.

Moreover, we can mechanistically explain the increased average distance between Au* and Ãu found in Fig. 3c based on our NAC electronic structure analysis of O*, Õ, and the Au atoms in Fig. 4d and e as follows. The two critical gold atoms Au* and Ãu, which anchor OH⁻ in

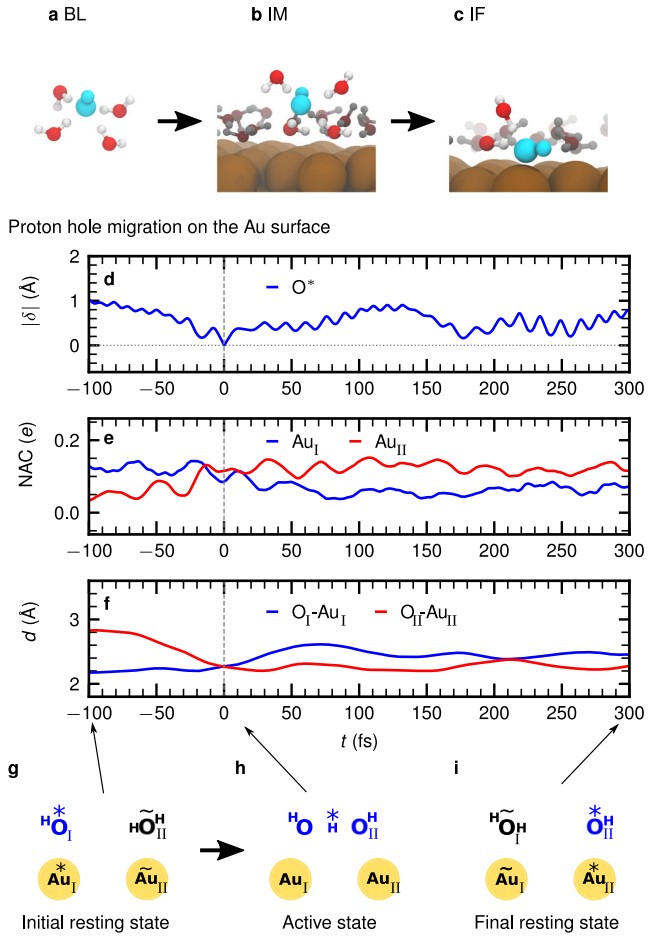

**Fig. 5 | Charge transfer mechanism at the alkaline gold/water interface via the OH⁻ charge defect.** The representative snapshots in (**a**–**c**) illustrate the solvation structure of the OH⁻ charge defect in the **a** bulk like (BL), **b** intermediate (IM) and **c** interfacial (IF) regions. In **a**, the water molecules which are not associated with the hypercoordinated first solvation shell (see text) of OH⁻ (cyan) are not presented while **b** and **c** include water molecules (gray) beyond the first solvation shell (oxygen in red). Representative time series referenced to the charge transfer event at $t$ = 0 fs of **d** the charge transfer coordinate $|\delta|$ involving the respective O*, **e** the NAC of the involved $Au_I$ and $Au_{II}$ atoms, and **f** the respective $O_I$-$Au_I$ and $O_{II}$-$Au_{II}$ distances $d$; see panels g--i for corresponding atom labeling. **g**–**i** Schematic illustration of the interfacial charge transfer mechanism of OH⁻ from the initial resting state via the active state to the final resting state anchored on the Au surface, see text, where the arrows indicate the corresponding times in panel f.

the resting state at Au* (see Fig. 4f–i) and together provide the bridge to hold the HB for charge transfer in the active state of OH⁻ at the gold surface (recall Fig. 3g), are both seen to gain positive charge (and most of it actually at Au*) compared to all other first-layer Au atoms. It implies that the strongly negatively charged solvation structures of OH⁻ (see Fig. 4b) locally induce the positive NACs of the gold atoms, which in turn repel each other which results in the observed increased distance between Au* and Ãu compared to all other surface Au pairs as shown in Fig. 3c.

**Distinct charge transfer mechanism of OH⁻ at the interface**
Based on our detailed analyses of structural and electronic properties of OH⁻ at the Au surface, we can now track the Grotthuss pathway of the hydroxide from bulk toward the electrified gold/water interface, Initially, OH⁻ resides in the BL region as visualized in Fig. 1b where it is in its favored hypercoordinated solvation state[15–17], see Fig. 5a. It then reaches the IM region it still retains its BL solvation pattern as depicted

in Fig. 5b and supported by the analysis presented in Supplementary Table 2. Before diving into the first solvation layer right at the Au surface, OH⁻ diffuses within the second water layer (see Supplementary Fig. 9). Finally, proton transfer from a water molecule in the first solvation layer to OH⁻ in the second layer displaces OH⁻ into the first layer via a structural diffusion event such that OH⁻ reaches the Au surface (Fig. 5c).

In the initial resting state within the first solvation layer on the Au surface, OH⁻ is solvated by several water molecules and sits on top of the Au* atom (labeled here as $Au_I$) as schematically depicted in Fig. 5g; note that we introduce for clarity in this figure the additional notation $Au_I$ and $Au_{II}$ as well as $O_I$ and $O_{II}$ in the analysis in panels e and f as defined in panels g–i therein. In this resting state of OH⁻, the thermal motion of water molecules leads to fluctuations of the charge transfer coordinate $|\delta|$ around 1.0 Å which is far from any charge transfer event (see Fig. 5d at $t$ = −100 fs before the charge transfer event at $t$ = 0 fs). When the NAC of $Au_{II}$ (in Fig. 5e) increases toward the value of $Au_I$, which occurs around $|\delta| \approx 0$ Å where the charge state of these two gold atoms starts to get exchanged, the most active proton, H*, is observed to transfer from $O_{II}$ to $O_I$ (see Fig. 5g toward h) as monitored by the decreasing value of $|\delta|$ for $t \to 0$ fs in panel d. Concurrently with this proton transfer, the proton donating $O_{II}$, which is the nascent OH⁻ after this charge transfer, approaches the $Au_{II}$ site to stabilize itself (see Fig. 5f). At the very charge transfer event, the active H* is centered between the two oxygen atoms $O_I$ and $O_{II}$ of the active OH⁻ state, i.e., close to the midpoint of the corresponding HB as visualized in Fig. 5h, implying $|\delta| \approx 0$ Å in Fig. 5d. As described in Fig. 3g, the two Au atoms $Au_I$/$Au_{II}$ and the two oxygens $O_I$/$O_{II}$ of the active OH⁻ complex form the pillar-and-bridge structure with the active H* that establishes the centered HB between $O_I$ and $O_{II}$ which characterizes the active state sketched in panel h. At this point, the positive partial charges of $Au_I$ and $Au_{II}$ become identical (see Fig. 5e at $|\delta| \approx 0$ Å). After that charge transfer event, the OH⁻ complex and the two supporting pillar atoms $Au_I$ and $Au_{II}$ relax into a new resting state that is translocated by one HB, corresponding to about one Au–Au lattice distance on the Au(111) surface, Fig. 5i, with respect to the original one shown in panel g. Correspondingly, in the final resting state, the NACs of the two anchoring gold atoms are reversed compared to the initial resting state depicted in panel g (see Fig. 5e).

## Discussion
Our results show that polarization and charge transfer effects between the gold surface and water molecules significantly affect the propensity of $H_3O^+$ and OH⁻ close to the Au surface. The hydrated OH⁻ complex is found to be strongly anchored at Au atoms and features peculiar solvation structures at the Au surface where water molecules in the hypercoordinated state, well-known from bulk water, get replaced by gold atoms. Thus, we find that OH⁻ strongly interacts with the electrified gold electrodes in our simplified computational model. This is qualitatively in line for instance, with Raman spectroscopy and other experimental findings, which demonstrate that OH⁻ ions can strongly adsorb onto Au surfaces within the double layer – even at pzc conditions – as clearly evidenced by a distinct Au–OH⁻ vibrational mode[45–49]. Transcending such previous insights, we now provide the full atomistic picture of how OH⁻ can anchor itself on gold surfaces despite its pronounced hypercoordinated solvation structure, and how such an anchored charge defect can possibly undergo Grotthuss charge migration within the IF region. Moreover, experimental studies showed that the potential of zero free charge (pzfc) shifts to negative potentials at Pt electrodes in alkaline solutions[2,50,51]. Within our simplified computational setup, the NACs of the specific anchoring gold atoms, Au* and Ãu, imply that they are polarized oppositely with respect to the attached OH⁻ ion. This suggests a qualitative trend, namely that higher pH values, leading to higher concentrations of OH⁻

species within the IF region, will induce larger negative pzfc values in accord with experimental observations[50,51].

When OH⁻ is in the bulk, it maintains a stable solvation structure with 4–5 water molecules (see Fig. 5a). However, the charge transfer to the first solvation layer requires the desolvation and reorientation of the water molecules (as illustrated in Fig. 5b) to form the specific solvation structures of OH⁻ at the topmost Au layer (see Fig. 5c as well as Fig. 4f–i). An experimental study[2] has suggested that the strong electric field generated by the induced surface charge aligns the water molecules at the interface, leading to a large reorganization energy required for water to transfer OH⁻/H₃O⁺ within the double layer. In addition, Fig. 4a suggests that the preferential location of OH⁻ near Au interfaces is closely related to the excess charge of water normal to the electrode. Specifically, OH⁻ in the BL region must overcome the oscillatory barriers to reach the first water layer. Therefore, we propose that the hindered transport of OH⁻ into the first solvation layer is due to (1) the transformation of the hypercoordinated solvation structure of OH⁻ from bulk to that involving surface Au atoms instead of water molecules and (2) the oscillatory excess charge within water near the Au surface in concert with hosting the new solvation structure of this charge defect that requires specific H-bonding.

In the case of H₃O⁺, our results show that the first water layer at the Au surface is not favorable to be populated by H₃O⁺ species even at negatively charged surfaces. It is known that Au has a positive hydrogen binding energy, which is often related to the HER reaction rate[24,52–54]. Together with the results of the excess charge profiles normal to the electrode (Fig. 4), our computational model suggests that the slow HER reaction at the Au electrode might be qualitatively attributed to not only the positive hydrogen binding energy, but also to the oscillatory feature of the NAC at the interface that disfavors H₃O⁺ ions to localize in the first layer. Importantly, we note that it has been reported that H₃O⁺ can integrate well into bilayer water structures[18], indicating that the solvation structure alone cannot fully explain the preferential location of this ion, yet the defect must be able to form HBs to be hosted. Additionally, a previous study employing STM and DFT techniques[55] found that H₃O⁺ resides predominantly in the second layer. These findings collectively suggest that the oscillatory NAC profile plays a pivotal role at liquid/metal interfaces together with proper solvation of the specific charge defects within the local HB network of interfacial water.

Assuming that one of the hydrogen atoms in the H₃O⁺ species is pointing to metal surfaces, the probability of hydrogen adsorption on the surface becomes higher as invoked in some previous work. However, our results indicate that the hydrogen atoms of H₃O⁺ species tend to form HB bonds with surrounding water molecules, rather than interacting with the Au surface. This arrangement of H₃O⁺ on gold in conjunction with the excess charge profile of water normal to the electrode surface that disfavors this charge defect to localize in the first solvation layer, greatly reduces the probability of hydrogen atoms of H₃O⁺ to adsorb on the Au surface. This computational finding might be qualitatively related to the observed imperceptible underpotential deposition of hydrogen for Au electrodes[1,56,57].

In summary, we utilized extensive AIMD simulations of gold/water interfaces carried out a zero and several finite surface charge densities to reveal in detail the solvation structures of OH⁻ and H₃O⁺ species at electrified gold electrodes that are in contact with water. In particular, our results indicate that the electronic structure of solvent molecules strongly affects the structure of OH⁻ and H₃O⁺ at the electrified Au interface. Specifically, OH⁻ prefers to anchor on top of some gold atom of the surface, Au*, which is positively polarized in contrast to the attached OH⁻ species. This positively charged Au* also serves as a donor to establish a hypercoordinated solvation structure of OH⁻ at the Au surface by replacing hydrogen-bonded water, as known from hypercoordination in bulk. This peculiar "resting state" of the hydrated hydroxide is unknown from bulk-like environments but characteristic

to OH⁻ at electrified gold/water interfaces. Together with a nearest neighbor atom, Ãu, these two gold atoms act as pillars to support the O*–H*–Õ bridge that enables charge transfer within the first solvation layer. Also, this "active state" of OH⁻ in interfacial water is distinctly different from active states seen before. These specific states of OH⁻ at the gold/water interface imply a unique Grotthuss-like mechanism that critically involves gold atoms from the electrode. This mechanism leads to a significant increase of the charge transfer barrier, which enormously slows down charge transfer via hydroxide defects in the first water layer on gold. In stark contrast to OH⁻, the solvation structures of H₃O⁺ at the gold/water interface are found to be very similar to those in the bulk liquid, leading to the same barriers to charge transfer via hydrated protons. Moreover, H₃O⁺ species are not favored in the first water layer both at zero and finite surface charge density conditions, which can be explained by the positive excess charge of water in the first solvation layer very close to the gold electrode.

Based on our findings, we suggest that the characteristic excess charge modulation of water normal to the metal electrode most likely plays a significant role in electrochemical processes more generally.

## Methods

The AIMD simulations[14] have been carried out at different controlled surface charge densities using the freely available CP2K code (version 8.0 and 10.0) with its Quickstep module[35–37] with a time step of 0.5 fs in the canonical (NVT) ensemble at 300 K. The underlying methodology based on the computational neon counter electrode[34] has been implemented by us in CP2K which allows us to apply finite computational potential bias during these AIMD simulations that controls the resulting surface charge density as illustrated in Supplementary Fig. 2, while not providing access to proper double-layer capacitance due to the deliberate lack of electrolyte ions in water. The temperature was controlled by thermostatting separately the water molecules and the mobile Au atoms using two independent Nosé-Hoover chain thermostats, see below. The RPBE density functional[58] with the D3 dispersion correction[59] was used to describe the aqueous HClO₄ and NaOH systems in view of the good performance of RPBE-D3 to describe water[60] and electrochemical electrode/electrolyte systems[61–64] with affordable computational cost in the framework of extensive AIMD investigations. It is well established that different density functionals can lead to incorrect structural diffusion scenarios for the charge self-defects in water[65,66] as reviewed in refs. [17,67]. The RPBE functional with D3 dispersion corrections as used in this study has been shown recently to provide accurate estimates of the diffusion coefficients for excess protons and hydroxide ions in water, producing a $D(H_3O^+)/D(OH^-)$ ratio that matches the experimental value[68]. We note in passing that more efficient machine learning approaches to simulate electrochemical cells at finite bias as required here, are yet under development. Following previous validation, the D3 correction was applied to all water molecules and to the topmost layer of Au atoms[61–64]. The core electrons were represented by GTH pseudopotentials[69,70], and the triple-ζ plus double polarization Gaussian basis sets TZV2P[71] were used. The auxiliary plane wave cut-off to represent the electronic density was set to 400 Ry using the Γ-point.

The dimensions of the computational supercell was set to $(48.57 \times 15.667 \times 15.076)$ Å³ subject to three-dimensional periodic boundary conditions (where the long axis is $x$ according to CP2K setup but defines the perpendicular axis $z$ normal to the Au surface in the presented graphs). The supercells hosting the aqueous HClO₄ and NaOH solutions (where the acid and base species are fully dissociated resulting in ClO₄⁻ and Na⁺ counterions close to the neon counter electrode) consist of a $(5 \times 6)$ periodic Au slab with 4 layers (with 30 atoms in each layer where the atoms in the bottom Au layer have been fixed at their lattice positions) to represent the gold electrode, one coplanar monolayer of 30 fixed neon atoms to represent the computational counter electrode[34], as well as 198 and 199 explicit water

molecules to solvate the fully dissociated $HClO_4$ and NaOH species, respectively.

The alkaline system was initially prepared by taking out one water molecule from our original equilibrated pure water 120 Au/200 water/ 30 Ne supercell while inserting the $Na^+$ and $OH^-$ ions. In case of the acidic system, the $ClO_4^-$, $H_3O^+$ and water molecules were randomly inserted between the Au slab and the Ne layer. Subsequent force field MD simulations were carried out to relax and pre-equilibrate at zero bias followed by the AIMD simulations to finally equilibrate and generate the trajectories for analyses. The bottom layer of the Au slab and the Ne monolayer were fixed during all simulations as to establish a water density of $\approx 1\,kg\,L^{-1}$ in the central region normal to the surfaces in an effort to represent bulk-like water within the same setup as interfacial water. A vacuum region with an extent of 12 Å (along the periodic $z$ axis) was introduced between the bottom layer of the Au slab and the Ne monolayer. The usual dipole correction was applied between the two electrodes within the vacuum region as required in the potentiostat method[34].

Overall, our systems are large compared to standard setups used in previous AIMD studies of electrified metal/water interfaces in the framework of explicit all-atom approaches. Our aims were to provide sufficient lateral space to allow us to simulate and study migration pathways of individual $H_3O^+$ and $OH^-$ charge defects in the interfacial region close to the metal electrode thus avoiding any interference effects due to other such defects, to also separate $H_3O^+$ and $OH^-$ from their counterions through explicit water, and to provide bulk-like water in the central region normal to the electrode surface along the $z$-axis as much as feasible. In practice, the pH of such solutions can influence the surface properties, whereas our simulation setup is able to represent the intrinsic solvation structures of the individual $H_3O^+$ and $OH^-$ charge defects in acidic and alkaline gold/water interfaces but not the impact of concentration effects and additional electrolyte ions.

The production runs for the acidic and alkaline electrolytes at finite surface charge densities were approximately 380 ps (from 16 independent trajectories) and 230 ps (from 12 independent trajectories), respectively, after careful equilibration periods. The corresponding systems at zero surface charge density (pzc) were propagated for approximately 150 ps (from 9 independent trajectories) and 350 ps (from 17 independent trajectories), respectively. Several trajectories at pzc condition with a total length of $\approx 170$ ps were used to compute the RDFs, surface/O-H angles and free energy profiles of $H_3O^+$ in the first solvation layer (generated by applying a quadratic mechanical wall potential setting in at $z_O = 4.3$ Å to act on the $z$-position of the $H_3O^+$ charge defect, determined using the collective variable from Supplementary Note 2 in order to restrain the migrating defect to stay within the IF region). The presented results are thus based on a grand total of 1.3 ns of AIMD sampling.

The number of configurations used to compute the NACs of O and H atoms in Fig. 4a were approximately 700 and 500 frames for the acidic and alkaline aqueous solutions, respectively. For the neat water, 100 equidistant configurations from 10 independent AIMD trajectories, each with a length of 10 ps, were used to compute the NACs of O and H atoms. Ten frames were used to compute Fig. 4b and 1700 frames were used to produce Fig. 4d and e. We used the DDEC partial charge code[43,72] to compute the NACs. We analyzed the HBs with a geometric definition[73]

$$r_{H\cdots O} < -1.71\,\text{Å}\cos\theta_{O-H\cdots O} + 1.37\,\text{Å}, \quad (1)$$

where $r_{H\cdots O}$ and $\theta_{O-H\cdots O}$ are the intermolecular distance O $\cdots$ H and the angle between the intermolecular O $\cdots$ H and intramolecular O−H vectors. Customized MDAnalysis code[74] (version 2.0.0) and custom Python code[75] (version 1.19.0) were used to analyze the trajectories. VMD software (version 1.9.3) was used to visualize and capture the snapshots[76].

## Reporting summary

Further information on research design is available in the Nature Portfolio Reporting Summary linked to this article.

## Data availability

Source data are provided with this paper. The data have also been deposited at https://doi.org/10.17877/RESOLV-2025-MFFC7VM3. All other data supporting the findings of this study are available within the paper and its supplementary information file or from the corresponding author upon reasonable request.

## Code availability

The codes used in this study are available from the corresponding author upon reasonable request.

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

## Acknowledgements

The authors thank Jörg Neugebauer and his group for helpful discus-sions on their potentiostat method and Lisa Hetzel for help with some preliminary work at an early stage. Funded by the Deutsche For-schungsgemeinschaft (DFG, German Research Foundation) under Ger-many's Excellence Strategy – EXC 2033-390677874 – RESOLV. Computing resources have been provided by HPC@ZEMOS, HPC-RESOLV, and BOVILAB@RUB. S.G. would like to thank the Department of Atomic Energy, Government of India, under Project Identification No. RTI 4007, for funding in India.

## Author contributions

D.M. conceived and designed the project. C.P., S.G. and H.F. imple-mented novel methodologies in the CP2K program package. C.P. per-formed the AIMD simulations, collated the data, conducted the analyses and wrote the first draft of the manuscript. All authors thoroughly dis-cussed the results and revised the manuscript.

## Funding

## Competing interests

The authors declare no competing interests.
