## [Transparent Peer Review file · Nature Communications]

Distinct solvation patterns of OH⁻ versus H₃O⁺ charge defects at electrified gold/water interfaces govern their properties

Corresponding Author: Dr Chanbum Park

Version 0:

Reviewer comments:

Reviewer #1

(Remarks to the Author)

The paper by Park et al presents a computational study of the Au/water interface under bias potential. The authors put effort on achieving large atomistic models and on producing several trajectories for each model. This allows to go beyond the limitations of small atomistic models and achieve good statistical sampling. The finding of this work are interesting, particularly those explaining the role of Au atoms in stabilizing the OH⁻ defect at the electrode surface. Therefore, this work might be worth publishing in Nat. Comm. journal given that the authors properly address the following issues:

The main issue with this work is the constant bias potential method used by the authors that is strongly remindful of a capacitor design rather than an electrochemical cell! This involves different charge dynamics, electric field strengths and consequently should have an impact on the electrochemical reactions at the vicinity of the active electrode. Actually, real systems should experience capacitance-related effects and non-Faradaic processes (e.g. charging of the double layer). I am not sure that these effects are properly modeled by Neugebauer's technique.

In addition, the authors find that there is no potential effect on the H₃O⁺ and OH⁻ reorientation which means that the interface is subject to a very weak electric field. Reorientation of H₃O⁺ and OH⁻ as a function of bias potential was already observed on several other metal/water interfaces.

Furthermore, an important issue is that real value of the applied potential is not known as it is not referenced to reference electrode (RHE, SHE). This is inherent to the applied methodology that lacks an external alignment level.

Therefore, the actual strength of the potential is ambiguous. -3V and +1V are very strong potentials compared to potential ranges applied experimentally, nevertheless the authors tend to say that they have minor effects on the defect orientation. Would it be possible to have extra runs at various applied potentials to have a complete picture of the potential range effects on the defects and EDL structures?

Finally, before going into the structure of defects at the interface, it would be of paramount importance to investigate the EDL evolution (H₂O orientation, HB network etc) as a function of the applied potential as it has been already done by several authors on Pt(111)?

A comparison would be of interest.

This authors are invited to address all the aforementioned issues to lift ambiguity and provide a solid ground to their investigation.

Reviewer #2

(Remarks to the Author)

The authors investigated the OH⁻ and H₃O⁺ solvation structures at electrified Au/water interfaces using AIMD simulations, and found adsorption complex structure of OH⁻ at the Au interface. While the AIMD simulations offered some atomistic detail on the interface structure, the connection to electrochemistry is obscure. It is strongly recommended that the authors should compare and discuss their work in the context of existing literature, particularly on electrochemical experiments. In its current form, the paper may not meet the standard for publication in Nature Communications.

:
Major points:

1. The statement that “Au* acts like a hydrogen-donating species that stabilizes OH⁻ at the Au surface together with the solvating water molecules” requires further experimental validation. Numerous benchmark cyclic voltammetry studies on Au/aqueous interfaces have demonstrated the presence of OH⁻ adsorption regions at certain potentials (Catalysis Today 262 (2016) 41–47). Providing additional experimental support or addressing the discrepancy with previous works could strengthen their claim.
2. While the authors' findings on the solvation structures of anchored OH⁻ species are interesting, drawing a direct connection to the sluggish kinetics of the oxygen evolution reaction (OER) requires further justification. The interfacial structures in these two systems are different, and the evidence presented may not be sufficient to establish a causal relationship. It would be advisable for the authors to clarify the distinctions between the two systems and provide additional experimental or theoretical support for their proposed connection. Without addressing these disparities, the conclusion regarding the impact on OER kinetics appears premature.
3. In addition, the electrode potential obviously goes beyond the scope of the stable potential range of metal/aqueous interfaces. For example, many reactions can take place at the extremely negative potential of -3V instead of maintaining a non-reactive metal/aqueous interface to study the structure of OH⁻ and H₃O⁺.
4. In Figure 2 b, it seems that the angle distribution of the Zundel ions has not been converged? Why there are so many spikes?
5. In Figure 2 c, it seems that the angle of OH⁻ moves from 78° at PZC to 84° at 1V. While the author stated that “OH⁻ is essentially parallel to the surface irrespective of the applied bias”. In addition, it is strange that the OH⁻ tends to be parallel to the surface as the potential is raised positively instead of the reverse orientation.
6. The authors stated that “the radial distribution functions (RDFs) and free energy profiles for charge transfer in Fig. 3a and d show that the properties of the excess proton within the IF region at the gold/water interface remain essentially unchanged with respect to those in the BL region (BL as defined in Fig. 1).” The author should provide the RDFs of OO for water molecules at the IF and BL region and make a comparison. Then make a comparison between the RDFs of H₃O⁺-O and H₂O-H₂O. After all, the proton transfer process will constantly occur. If the authors use the fixed index to determine the RDF but the proton has transferred to another water molecule, then the intrinsic RDF of H₃O⁺-H₂O will be hidden by the RDF of H₂O-H₂O due to its large amount at the interface.
7. Following the previous question, it would be better if the author could analyze RDF of Zundel and Eigen ions at different conditions separately to determine if the bias will affect solvation structures of these ions.
8. The authors stated that “Thus, we find that OH⁻ strongly interacts with electrified gold electrodes which is in line for instance with Raman spectroscopy finding that OH⁻ ions adsorb on Au surfaces.” According to the Raman spectroscopy (pH ~9), the adsorbed OH is present at the potential range 0.40-1.0 V vs. Ag/AgCl. It is not clear what the pH condition the model corresponds to although the electrode potential has been reported. A more careful comparison is needed.

Minor points:

1. Previous studies suggest that some density functionals (BLYP, PW91, HCTH) would yield qualitatively different solvation structures, diffusion constants. This manuscript should at least comment the possible effect of the functionals to the solvation structures. After all, there is no benchmark comparison to the experimental results during the study of the interfacial structures.
2. I sometime got confused by the abbreviations used in this manuscript. For example, the IF for interface (also BL and IM). It would be better if they follow the tradition as those used in the study of metal/water interfaces.
3. For Figure 2 d-f, I think it will be better if they highlight the key structures and represent all other water molecules with smaller balls and sticks or just use sticks.
4. The presentation of the density profiles in Figure 1 d and e are also not easy to read. They can delete the density distribution of the metals and take the top of the Au surface atoms as reference plane.
5. It is also not clearly stated which surface is studied in the method section. And it takes me a while to realize that the author is studying Au(111) at line 568.

Reviewer #3

(Remarks to the Author)

The manuscript “Distinct solvation patterns of OH⁻ versus H₃O⁺ charge defects at electrified gold/water interfaces govern their properties” presents an ab initio molecular dynamics (AIMD) study of the interface between charged gold surfaces and liquid water. The specific focus of the manuscript is the interfacial solvation of hydroxide and hydronium species and how solvation is affected by applied electrode potentials. The simulations reveal that OH⁻ and H₃O⁺ are solvated differently at the interface, with OH⁻ tending to reside in close contact with the gold surface and H₃O⁺ tending to reside further away, approximately within the second solvent layer. These results and this contrast are interesting and may provide insight into the longstanding debate about the interfacial solvation properties of water ions. However, the explanation provided for these differences, i.e., that they reflect the response of ions to water's underlying net atomic charge (NAC) profile, is not adequately supported by the data. Indeed, there appears to be a correlation between the positive regions of the NAC and the preferred location of OH⁻, and (to a lesser extent) a correlation between the negative regions of the NAC and the preferred location of

H3O+. Whether this correlation is causal or incidental is not established. For this reason, I cannot recommend publication in Nature Communication. Below is a list of issues that the authors should consider upon revision.

1. Part of this study focuses on the influence of applied potential on the solvation properties of the water ions. The authors consider four conditions, pzc with OH⁻, pzc with H3O⁺, -3V with H3O⁺, and +1V with OH⁻. I find it odd that the applied potential values are different for the hydroxide and hydronium case. Why did the authors not bias both at +1V or +3V? Was -1V insufficient to draw H3O⁺ towards the interface?
2. I think it would be useful to see a comparison of the unbiased and biased OH⁻ and H3O⁺ distributions (e.g., in Figs. 1d and 1e). This would highlight an interesting contrast in the responsiveness of interfacial H3O⁺ vs OH⁻ to applied potential.
3. The definition of the CV for water ion position as presented in the SI is confusing to me. If I understand things correctly, then $n_i=2$ for all water molecules, so the summation in Eq. 2 would include many terms of $z_i \cdot \exp(+40)$ and approximately 1 term with (hydronium $n_i=3$) $z_i \cdot \exp(60)$ or (hydroxide $n_i = 1$) $z_i \cdot \exp(-20)$. I would expect the terms in the summation to look something like $z_i [1 - \exp(-\lambda \cdot (n_i - 2))]$.
4. I find it very surprising that at least one of the hydrogens on H3O⁺ do not orient towards the electrode (like they do for water). Is there a precedent for this finding? Can the electronic structure calculation help to rationalize this finding? Maybe the shape of the MOs provide some insight?
5. The authors state on page 5 that "Clearly, the positive potential bias drives the OH⁻ defect into the first water layer,...". Based on the results at pzc, this statement is misleading. The OH⁻ appears to be driven to the first water layer even without a positive potential bias.
6. If the charge δq oscillations for water drive the interfacial solvation properties of these species, then I would expect the density profiles in Fig. 1d and 1e for OH⁻ and H3O⁺ to roughly mirror the oscillations of δq . For instance, if region of positive NAC is responsible for the solvation position of OH⁻, then why isn't there a secondary OH⁻ peak around $z=12$, where there is another positive NAC region?
7. In the discussion section, I think claiming that the oscillatory feature of the NAC "repels H3O⁺ ions from the first layer as we demonstrate" is much too strong of a statement. In fact, I'm sure much of the difference between the H3O⁺ and OH⁻ in these systems has to do with the solvation (or desolvation) free energetics.

Version 1:

Reviewer comments:

Reviewer #1

(Remarks to the Author)

While the authors agree with my comments on the limitations of the constant potential method, they tend to omit these very important aspects from the main manuscript. This work is a first application of Neugebauer's method after implementation in cp2k code. Cp2k is a widely used code and I am afraid if the authors do not include a clear word of caution about the limitations of the potentiostat method and specifically that it is not a method suitable for modelling electrochemical cells, this will induce unexperimented future users into error. Not everyone is aware of these limitations and most of users will tend to apply it blindly. Therefore, I invite the authors to add a clear and honest statement about the limitations of this technique in a future revision.

This method cannot fundamentally simulate correctly a bias-potential and its corresponding charge density at the electrode surface. This means that the bias potential reported by the authors does not correspond to the reported charge density (cannot predict capacitance!). Given this fundamental limitation, the authors suggest to simply use the surface charge density to define their systems. If this choice is kept, the paper should then be reshaped to reflect that these are not constant-potential simulations and not even a proper way to model electrochemical cells.

As such, this method can only help to access reaction mechanisms at various surface charge densities without quantitatively accessing the electrochemical properties of the interface in a way comparable to experiments. The conclusions must then be tuned accordingly.

Reviewer #2

(Remarks to the Author)

The authors have made efforts to address the comments raised. I am fine with most of them, but still not convinced of the argument on the unphysically too negative electrode potential. Previous AIMD work in literature can well reproduce the capacitance, i.e. relation between surface charge density and electrode potential. Why is there such a large discrepancy that leads to the very negative potential -3 V, estimated from the surface charge density. Also, it seems that at -3V the OH angle and RDF of the ions are not affected much compared to the PZC condition. I am very concerned on how the electrode potential is determined and how accurate the interfacial structures are.

A typo: At line 171, "as a has been realized using a continuous"

Reviewer #3

(Remarks to the Author)

The revisions and responses have addressed some of my previous concerns. The results highlight some interesting differences in the interfacial solvation of H_3O^+ and OH^- . However, I think the results lack the scientific novelty to warrant publication in Nature Communications. The primary conclusions, e.g., that OH^- coordinates to the gold surface, have an experimental precedent. Highlighting specific details of the solvation structure, e.g., orientational angles, and their subtle dependences on applied potential certainly merit publication, but these results do little to transform the way we think about the electrochemical processes that these species are involved in. Thus, I recommend the authors submit this manuscript to a more specialized journal.

Version 2:

Reviewer comments:

Reviewer #1

(Remarks to the Author)

The authors addressed all my concerns and provided a clear and detailed explanation of the limitations of their methods. The extensive computational efforts in delivering high-quality first-principles modelling and the novel mechanistic aspects discovered by the authors convinced me that this work is worth publication in Nature Communications journal.

Reviewer #2

(Remarks to the Author)

The authors have made further revisions to their original draft and most of my concerns have been addressed in this version. I would recommend its publication in Nature Communication provided that the following minor revisions are made.

1. The author should explicitly outline the potential issues associated with electrode potential calculations in this paper. This will prevent readers from mistakenly assuming that this method can be utilized for electrochemical studies without additional critical evaluation.
2. In Figure 3a and 3b, I recommend that the authors include RDF data for H_2O - H_2O as reference. This addition will allow readers to observe the relative peak position shifts surrounding H_3O^+ / OH^- .

1 REVIEWER COMMENTS

REVIEWER COMMENTS

Reviewer #1 (Remarks to the Author):

The paper by Park et al presents a computational study of the Au/water interface under bias potential. The authors put effort on achieving large atomistic models and on producing several trajectories for each model. This allows to go beyond the limitations of small atomistic models and achieve good statistical sampling. The findings of this work are interesting, particularly those explaining the role of Au atoms in stabilizing the OH^- defect at the electrode surface. Therefore, this work might be worth publishing in Nat. Comm. journal given that the authors properly address the following issues:

Response: We thank the reviewer very much for pointing out that we invested a lot of effort in simulating large atomistic models using several trajectories for each model, thus going beyond the usual small atomistic models and meager data statistics. As you will see, we took great care to address all issues and recommendations raised as follows.

The main issue with this work is the constant bias potential method used by the authors that is strongly remindful of a capacitor design rather than an electrochemical cell! This involves different charge dynamics, electric field strengths and consequently should have an impact on the electrochemical reactions at the vicinity of the active electrode. Actually, real systems should experience capacitance-related effects and non-Faradaic processes (e.g. charging of the double layer). I am not sure that these effects are properly modeled by Neugebauer’s technique.

Response: We of course fully agree with the reviewer that Neugebauer’s technique is based on a parallel plate capacitor model. Consequently, mimicking a full electrochemical cell setup would require a proper distribution of ions in addition which for sure will impact electrochemical reactions close to the electrode. However, we would like to emphasize that the purpose of our paper is not to realistically model the electrochemical environment, and even less to electrochemical reactions, but rather to investigate the impact of bias on the two very important water self-defects in electrolytes, namely hydronium and hydroxide, since such molecular insights are still largely missing at the full atomistic level.

In this context, we consider Neugebauer’s method to be ideal for the current purpose since it does not require a compensating background charge or additional counter ions to electrify the electrode as required in other computational approaches to model electrochemical setups. This feature makes the technique particularly suitable for exploring the *intrinsic properties* of H_3O^+ and OH^- at the Au surface under bias potentials in the absence of other ions close to the electrode which are known to *strongly perturb* the interfacial water structure specifically if still only rather small electrode surfaces are available which do not allow for proper decoupling of the respective solvation shells. In the same spirit, by applying our bias potentials we make sure to stay away from any electrochemical reactions which are known to occur beyond our maximum potentials for the two cases, see our response to the next question for justification, details and changes to the text.

To clarify our aims and thus our specific computational approach, we have revised the text in the manuscript as follows: “... we adapt a potentiostat method with an explicit computational neon counter electrode which allows for AIMD simulations under bias potential conditions;” as well as “... to unravel the intrinsic properties of the H_3O^+ and OH^- charge defects at gold electrodes in contact with water while applying bias potentials. Having this in mind, we stress that our maximum potentials (see below) are chosen such that we safely stay away from any interfering electrochemical reactions in both cases, and we also do not include any additional ions to not perturb the HB network of the interfacial water layers as a result of hosting their solvation shells.”

In addition, the authors find that there is no potential effect on the H_3O^+ and OH^- reorientation which means that the interface is subject to a very weak electric field. Reorientation of H_3O^+ and OH^- as a function of bias potential was already observed on several other metal/water interfaces.

Response: As stimulated by the first remark, we now correlate the surface charge densities obtained in our calculations (see new Supplementary Table 1 in the SI) with the experimental values reported in the literature [20, 4]. Importantly, this comparison to the charge densities reported versus the electrode potential in Revision Figure 1 demonstrates that our applied biases (electric fields) are in the regime that should not induce any electrochemical reaction which would strongly interfere with the intrinsic molecular properties of the H_3O^+ and OH^- charge defects close to the metal electrode; there is of course the unavoidable caveat that experiments provide macroscopic surface charge densities whereas the ones computed here are microscopic properties since based on fully atomistic simulations.

In response to this issue, we amended Supplementary Section 1 as follows: **“Based on the data in Supplementary Table 1, we can correlate the surface charge densities obtained from our calculations, based on the computational bias, with experimental values reported in the literature [20, 4]. Comparison to these experimental charge densities as a function of the electrode potential [20, 4] demonstrates that our computational voltages as defined above are in the regime that should not induce any electrochemical reaction in the acidic and alkaline cases. This is important since such reactions would strongly interfere with the intrinsic molecular properties of the H_3O^+ and OH^- charge defects close to the gold electrode on which we focus in this investigation.”**

Secondly, for computational methods where charging the simulation cell requires introducing excess ions in the electrolyte (ion unbalance approaches), the intrinsic solvation environment of the H_3O^+ and OH^- defects can be disrupted due to the presence of these excess ions with their own solvation shells, in particular when using small or moderately-sized AIMD supercells, allowing the orientations of the defects to align with the external field as a result of such perturbations of the hydrogen bond networks that stabilize the defects. Intrinsically, a well connected hydrogen bonding network formed by the water molecules parallel to the surface imparts stability to the hydronium towards field induced reorientation (in line with earlier findings in Ref. [22]).

Similarly, the OH^- defect is also constrained by its strong interaction with the surrounding water molecules as indicated by the donor number of OH^- at the interface (see Supplementary Table 2 in the SI). The hydrogen atom of OH^- is hydrogen-bonded to surrounding water molecules, forming an OH^- angle of approximately 80 degrees (see Figure 2c in the main text). Due to this integration and thus stabilization in the nearby hydrogen bond network of interfacial water, reorientation of the O–H angle along the electric field is quite constraint and is expected to occur only either when applying larger bias potentials (thus inducing electrochemical reactions) or due to the presence of other ions (thus breaking the hydrogen bond network in the first solvation layer).

We clarify this situation now in the revised manuscript by introducing a novel paragraph **“Overall, the structure of the hydrated excess proton in the IF region is found to be not much affected given the applied bias. Both, the Eigen- and Zundel-like solvation structures of H_3O^+ are found to be well-integrated in the HB network of the IF water layer which imparts orientational stability toward field-induced reorientation at the given bias potential. Similar conclusions will be reached for the hydrated proton hole defect as to the orientation of the OH vector with respect to the surface normal, see below. This general picture is expected to change when applying larger bias potentials (thus inducing electrochemical reactions) or due to the presence of other ions in the first solvation layer of the metal electrode (thus breaking the HB network herein).”** and by referring to additional analyses depending on bias **“... refined analyses of mass density profiles as well as the OH vector orientation is provided in Supplementary Fig. 12 for different bias potentials compared to pzc conditions).”**

Figure 1. (a) CV curves recorded at a Au(111) electrode in 0.1 M KClO_4 (dotted line) and in the supporting electrolyte upon the addition of KOH, 1.00×10^{-4} M, 1.00×10^{-3} M (dashed lines), and 1.00×10^{-2} M (solid line) at a sweep rate of 10 mV/s. (b) Charge density versus electrode potential curves for the Au(111) electrode in 0.1 M KClO_4 (dotted line) and 0.1 M $\text{KClO}_4 + x$ M KOH solutions.

Fig. 2. Charge density-electrode potential curves at the Au(111) electrode for the 0.1 M HClO_4 solution (\circ) with the following additions of K_2SO_4 : \bullet 5×10^{-6} M K_2SO_4 ; \square 2.5×10^{-5} M K_2SO_4 ; \blacksquare 5×10^{-5} M K_2SO_4 ; \triangle 10^{-4} M K_2SO_4 ; \blacktriangle 5×10^{-4} M K_2SO_4 ; \diamond 2.5×10^{-3} M K_2SO_4 ; \blacklozenge 5×10^{-3} M K_2SO_4 . For clarity only the data for a few concentrations are shown in the figure.

Revision Figure 1: Experimental surface charge density versus electrode potential curves for alkaline (left) and acidic (right) electrolytes in contact with Au(111) electrodes. Left and right figures taken from Refs. [4] (Reprinted with permission from Ref. [4]. Copyright 1999 American Chemical Society) and Ref.[20] (Reprinted from Ref.[20], Copyright (1994), with permission from Elsevier), respectively.

Concerning now the second suggestion: ... *Reorientation of H_3O^+ and OH^- as a function of bias potential was already observed on several other metal/water interfaces.*

To the best of our knowledge, the solvation environment of the proton defects in pure water in the vicinity of a charged Au surface has not been studied before; we compile related references below. To address this gap in the literature, we have included the references below in the revised manuscript and modified the text as follows: “ ... **for determining the reaction kinetics. Although there are studies that elucidate the behavior of H_3O^+ (see Refs. [16, 2, 19, 8, 7, 10]) and OH^- (see Ref. [5]) ions at metal surfaces, their intrinsic solvation structures remain unexplored at the atomistic level.**”

Furthermore, an important issue is that real value of the applied potential is not known as it is not referenced to reference electrode (RHE, SHE). This is inherent to the applied methodology that lacks an external alignment level.

Response: We are aware that converting computational potentials into reference potentials is not straightforward or even not possible at the rigorous quantitative level. For this reason, we reported the computational bias potential scale in the manuscript. Specifically, our computational potentials are referenced with respect to the potential of zero charge (pzc). The surface charge densities that are now reported in the new Supplementary Table 1 demonstrate the relevant bias range that can be compared to experiments; see our feedback to the next

point for details. **For clarity, we provide explanations about our computational potentials and the surface charge density corresponding to the computational voltages at the end of Section 1 of the Supplementary Information where also the new Table 1 is presented.** These additional data are important for answering the next question when it comes to compare our computational bias potentials to experimental electrode potentials.

Therefore, the actual strength of the potential is ambiguous. $-3V$ and $+1V$ are very strong potentials compared to potential ranges applied experimentally, nevertheless the authors tend to say that they have minor effects on the defect orientation. Would it be possible to have extra runs at various applied potentials to have a complete picture of the potential range effects on the defects and EDL structures? Finally, before going into the structure of defects at the interface, it would be of paramount importance to investigate the EDL evolution (H_2O orientation, HB network etc) as a function of the applied potential as it has been already done by several authors on $Pt(111)$? A comparison would be of interest.

Response: First of all, we refer back to our feedback on the previous question which clarifies how we define our computational potentials. We are aware that the computational potentials we report cannot be directly compared to experimental voltages, but the computed surface charge densities are more realistic and can be more directly related to the experimental values. This additional data are provided in the revised version of the Supplementary Information in the new Table 1. As one can see from that table, the surface charge densities in our AIMD simulations range from approximately -13 to $+4 \mu\text{C cm}^{-2}$. Upon comparison of these numbers to experimental data [20, 4], see the Revision Figure 1 from above, one finds that our underlying potential bias values correspond for both defects to the accessible voltage regime before any electrochemical reactions occur (as shown in these experimental publications), which we aim to avoid as explained previously.

Out of courtesy to the referee we call here our changes to the text in response to assessing the magnitude of our computational voltages compared to experiment: **“Based on the data in Supplementary Table 1, we can correlate the surface charge densities obtained from our calculations, based on the computational bias, with experimental values reported in the literature [20, 4]. Comparison to these experimental charge densities as a function of the electrode potential [20, 4] demonstrates that our computational voltages as defined above are in the regime that should not induce any electrochemical reaction in the acidic and alkaline cases. This is important since such reactions would strongly interfere with the intrinsic molecular properties of the H_3O^+ and OH^- charge defects close to the gold electrode on which we focus in this investigation”.**

Secondly, our findings based on additional analyses depending on bias potential (as already outlined earlier in our response) indicate that, in the regime before electrochemical reactions set in, bias potentials have only a minor effect on the defect orientations. This is due to the fully embedded solvation structures surrounding the defects within the strong hydrogen bond network provided by the interfacial water layers as discussed above. We again refer to our response already given above that is also based on additional analyses of orientational properties depending on applied potentials to have a complete picture, as requested by the referee: **“Overall, the structure of the hydrated excess proton in the IF region is found to be not much affected given the applied bias. Both, the Eigen- and Zundel-like solvation structures of H_3O^+ are found to be well-integrated in the HB network of the IF water layer which imparts orientational stability toward field-induced reorientation at the given bias potential. Similar conclusions will be reached for the hydrated proton hole defect as to the orientation of the OH vector with respect to the surface normal, see below. This general picture is expected to change when applying larger bias potentials (thus inducing electrochemical reactions) or due to the presence of other ions in the first solvation layer of the metal electrode (thus breaking the HB network herein).”** and by referring to additional analyses depending on bias **“... refined analyses of mass density profiles as well as the OH vector orientation is provided in Supplementary Figure 12 for different bias potentials compared**

to pzc conditions).”

Thirdly, we agree with the reviewer on the importance of studying the evolution of H₂O orientation and the HB network. **To address this, we have conducted additional analyses on the dependence of both H₂O orientation and the HB network on the bias potentials and included these results in the Supplementary Figure 4a and new Supplementary Figure 5.** It is clear from the corresponding figures that the HB network and H₂O orientation are sensitive to the value of the bias potentials compared to the pzc reference, reflecting the response of water orientations to the applied bias potentials. As one might have expected, the HB network statistics are identical in the bulk-like region, whereas they are clearly different in the interfacial water layer where we find that they depend on the bias potential.

Based on these additional analyses in response to the referee’s question, we now **extend panel a of Supplementary Figure 4 by adding new -2 V data, provide the new Supplementary Figure 5, and expanded the discussion in the main text as follows** “Simultaneously, the OH vectors of the IF water molecules predominantly point towards the Au surface at finite bias, resulting in **an increased number of dangling OH bonds compared to the pzc reference. This effect becomes more pronounced as the bias increases, reflecting the response of water orientations to the applied potential (see Supplementary Fig. 4a).** Moreover, detailed decomposition of the HB patterns of the individual water molecules in the IF layer unveils that the HB states also depend on bias conditions, whereas they are identical in the BL regime which features a distinctly different overall pattern dominated by molecules that accept and donate two HBs (D2A2 species), see Supplementary Fig. 5.”

This authors are invited to address all the aforementioned issues to lift ambiguity and provide a solid ground to their investigation.

We are very grateful for the careful report with many questions and suggestions which we all took very seriously since considering and implementing them based on additional analyses, comparisons to experiment and novel considerations greatly improved and actually supported even more the presented material and thus conclusions.

Reviewer #2 (Remarks to the Author):

The authors investigated the OH^- and H_3O^+ solvation structures at electrified Au/water interfaces using AIMD simulations, and found adsorption complex structure of OH^- at the Au interface. While the AIMD simulations offered some atomistic detail on the interface structure, the connection to electrochemistry is obscure. It is strongly recommended that the authors should compare and discuss their work in the context of existing literature, particularly on electrochemical experiments. In its current form, the paper may not meet the standard for publication in Nature Communications.:

Major points: 1. *The statement that “Au* acts like a hydrogen-donating species that stabilizes OH^- at the Au surface together with the solvating water molecules” requires further experimental validation. Numerous benchmark cyclic voltammetry studies on Au/aqueous interfaces have demonstrated the presence of OH^- adsorption regions at certain potentials (Catalysis Today 262 (2016) 41–47). Providing additional experimental support or addressing the discrepancy with previous works could strengthen their claim.*

Response: We thank the reviewer for raising the important issue regarding the adsorption of OH^- on gold. Several experimental studies [4, 1] including the reference suggested by the reviewer [12] have reported that OH^- adsorption can occur over a *wide range* of potentials – notably including even the potential of zero charge (pzc) limit. In particular, Figure 11 in the experimental study [4] provides a clear connection between the Gibbs excess of OH^- adsorption on the Au(111) surface and the O-H stretching mode of H_2O molecules hydrogen-bonded to the oxygen atom of OH^- including pzc conditions. Most importantly, an experiment using Shell-Isolated Nanoparticle-Enhanced Raman Spectroscopy [9] as well as other literature [11, 6] detected and assigned a very distinct Au-OH bending mode at $\sim 790\text{ cm}^{-1}$, which emerges prior to the complete oxidation of the Au surface. We mention in passing that using our thermostatted NVT trajectories, we did very preliminary VDOS analysis and found this peak at about 790 cm^{-1} in accordance with experiment.

These experimental references clearly indicate that OH^- can exist on gold over a wide range of bias potentials and, secondly, that it is strongly anchored on Au atoms, otherwise no Raman mode would be found, thus adding further credence to our results. Yet, no previous studies could provide the atomistic details on how exactly OH^- is anchored (given its strongly preferred hypercoordinated solvation structure in the bulk limit) and, more importantly, how such an anchored charge defect can possibly undergo Grotthuss charge migration within the first solvation layer which we provide here for the first time. **We conclude that there is no discrepancy with previous works, rather we provide the yet unknown atomistic insight into existing experimental support of stable Au-OH arrangements on metal surfaces.**

To address this topic, we modified the text as follows: “Thus, we find that OH^- strongly interacts with electrified gold electrodes which is in line for instance with Raman spectroscopy **and other experimental findings, which demonstrate that OH^- ions can strongly adsorb onto Au surfaces within the double layer – even at pzc conditions – as clearly evidenced by a distinct Au– OH^- vibrational mode.**^{4,11,6,9,12} Transcending such previous insights, we now provide the full atomistic picture of how OH^- can anchor itself on gold surfaces despite its pronounced hypercoordinated solvation structure, and how such an anchored charge defect can possibly undergo Grotthuss charge migration within the IF region.” where we provide additional references to the aforementioned works including the Catal. Today article.

2. *While the authors’ findings on the solvation structures of anchored OH^- species are interesting, drawing a direct connection to the sluggish kinetics of the oxygen evolution reaction (OER) requires further justification. The interfacial structures in these two systems are different, and the evidence presented may not be sufficient to establish a causal relationship. It would be advisable for the authors to clarify the distinctions between the two systems and provide additional experimental or theoretical support for their proposed connection. Without addressing these disparities, the conclusion regarding the impact on OER kinetics appears premature.*

Response: We clearly acknowledge that the OER involves multiple OH^- ions and a series of complex steps. A direct comparison between experimental results and our simulation study is thus beyond the scope of this work, but was merely intended as a suggestion to stimulate future work along the lines of our findings. In revising the manuscript, we sharpened throughout the focus of our work on disclosing the atomistic behavior of the OH^- and H_3O^+ charge defects at electrified gold electrodes. Therefore, we have removed the corresponding paragraph from the discussion section which is not vital given the focus of our study and other connections to experimental findings (for instance as worked out in response to point 1 above).

3. In addition, the electrode potential obviously goes beyond the scope of the stable potential range of metal/aqueous interfaces. For example, many reactions can take place at the extremely negative potential of -3V instead of maintaining a non-reactive metal/aqueous interface to study the structure of OH^- and H_3O^+ .

Response: We now understand that we were not clear enough in defining our *computational* bias potentials. **First of all, we explain the details on how we defined and computed the reported bias potentials at the end of Supplementary Section 1 and, moreover, we provide the corresponding surface charge densities in the new Table 1.**

Secondly, rigorously computing reference potentials at the quantitative level is difficult (or impossible) without adjustments whereas surface charge densities built a more reliable bridge between experiment and simulation. When comparing our surface charge densities from Supplementary Table 1 to experimental data [20, 4] in Revision Figure 2, see below, one can see that our computational voltages correspond to those experimental potential regimes, both for H_3O^+ and OH^- where no electrochemical reactions are expected to occur. There is of course the unavoidable caveat that experiments provide macroscopic surface charge densities whereas the ones computed here are microscopic properties since based on fully atomistic simulations. As now stressed better in the revised text, see above, our key objective is to investigate the impact of bias on these very charge defects in pure water in contact with gold electrodes, rather than electrolyte solutions, to unveil their atomistic properties. From this perspective, our simulation setup allows us to examine the intrinsic properties of H_3O^+ and OH^- at the Au surface under a bias potential – but in the absence of electrochemical reactions and strongly perturbing co-adsorbed ions.

To clarify the situation, we added the following analysis at the end of Supplementary Section 1: **“Based on the data in Supplementary Table 1, we can correlate the surface charge densities obtained from our calculations, based on the computational bias, with experimental values reported in the literature [20, 4]. Comparison to these experimental charge densities as a function of the electrode potential [20, 4] demonstrates that our computational voltages as defined above are in the regime that should not induce any electrochemical reaction in the acidic and alkaline cases. This is important since such reactions would strongly interfere with the intrinsic molecular properties of the H_3O^+ and OH^- charge defects close to the gold electrode on which we focus in this investigation”.**

Figure 1. (a) CV curves recorded at a Au(111) electrode in 0.1 M KClO_4 (dotted line) and in the supporting electrolyte upon the addition of KOH, 1.00×10^{-4} M, 1.00×10^{-3} M (dashed lines), and 1.00×10^{-2} M (solid line) at a sweep rate of 10 mV/s. (b) Charge density versus electrode potential curves for the Au(111) electrode in 0.1 M KClO_4 (dotted line) and 0.1 M $\text{KClO}_4 + x$ M KOH solutions.

Fig. 2. Charge density–electrode potential curves at the Au(111) electrode for the 0.1 M HClO_4 solution (○) with the following additions of K_2SO_4 : ● 5×10^{-6} M K_2SO_4 ; □ 2.5×10^{-5} M K_2SO_4 ; ■ 5×10^{-5} M K_2SO_4 ; △ 10^{-4} M K_2SO_4 ; ▲ 5×10^{-4} M K_2SO_4 ; ◇ 2.5×10^{-3} M K_2SO_4 ; ◆ 5×10^{-3} M K_2SO_4 . For clarity only the data for a few concentrations are shown in the figure.

Revision Figure 2: Experimental surface charge density versus electrode potential curves for alkaline (left) and acidic (right) electrolytes in contact with Au(111) electrodes (this figure is identical to Revision Figure 1 and reproduced here out of courtesy to the referee). Left and right figures taken from Refs. [4] (Reprinted with permission from Ref. [4]. Copyright 1999 American Chemical Society) and Ref.[20] (Reprinted from Ref.[20], Copyright (1994), with permission from Elsevier), respectively.

4. In Figure 2b, it seems that the angle distribution of the Zundel ions has not been converged? Why there are so many spikes?

Response: Stimulated by this concern of the referee, we increased the sampling by running additional trajectories at -3 V. Accordingly, **we have updated Figure 2 in the main text using the enhanced statistics.** The conclusion is that the improved data clearly strengthen our previous finding that the angle of the Zundel complex is close to 90 degrees.

5. In Figure 2c, it seems that the angle of OH^- moves from 78° at PZC to 84° at 1V. While the author stated that “ OH^- is essentially parallel to the surface irrespective of the applied bias”. In addition, it is strange that the OH^- tends to be parallel to the surface as the potential is raised positively instead of the reverse orientation.

Response: OH^- interacts with the Au atom through one of the lone pairs of its oxygen atom. Simultaneously, OH^- is strongly embedded within the solvation shells formed by surrounding water molecules and the Au^* atom to stabilize itself in the preferred hypercoordinated state. The donor number of OH^- within the interfacial region (Supplementary Table 2) indicates that the proton of OH^- is hydrogen-bonded to the water molecules at the first layer. Under these conditions, the OH^- ion does not align along the electric field direction given our bias potential (see discussion above in the context of surface charge densities). Consequently, the OH^- ion prefers an essentially parallel alignment w.r.t. the Au surface (considering in the original manuscript 80 degrees to be close enough to 90 degrees to use the term “parallel” in a qualitative context).

In response to the reviewer’s comment, **we have revised the statement cited above to remove any ambiguity in the main text as follows:** “... is roughly aligned with an O–H angle of approximately 80 degrees with respect to the surface at pzc conditions as well as at 1 V bias ...” and moreover we now provide new discussion of the comparison to the different orientation at mineral and graphene surfaces: **“At this point, we anticipate a key difference of OH^- in contact with the gold surface: We show below that OH^- strongly interacts with individual Au atoms (namely Au^* and $\tilde{\text{Au}}$, as analyzed below, through one of the lone pairs of its oxygen atom) in concert with establishing significant H-bonding to several surrounding water molecules to stabilize itself in the preferred hypercoordinated state while including gold atoms.”**

6. The authors stated that “the radial distribution functions (RDFs) and free energy profiles for charge transfer in Fig. 3a and d show that the properties of the excess proton within the IF region at the gold/water interface remain essentially unchanged with respect to those in the BL region (BL as defined in Fig. 1).” The author should provide the RDFs of OO for water molecules at the IF and BL region and make a comparison. Then make a comparison between the RDFs of H_3O^+ -O and H_2O - H_2O . After all, the proton transfer process will constantly occur. If the authors use the fixed index to determine the RDF but the proton has transferred to another water molecule, then the intrinsic RDF of H_3O_2 - H_2O will be hidden by the RDF of H_2O - H_2O due to its large amount at the interface.

Response: In response to the reviewer’s suggestion, we have now computed all relevant RDFs which we fully split into Eigen and Zundel states as well as interfacial and bulk-like regions for finite versus zero bias and now depict all those in the new Supplementary Figure 7. This additional analysis clearly supports our earlier finding that the solvation of H_3O^+ exhibits an almost identical solvation structure in the interfacial region, whether under finite bias or not, compared to bulk. To support our earlier statement based on this refined analysis, we added to the main text: **“see Supplementary Figure 7 for the comprehensive set of RDFs split into Eigen and Zundel states as well as IF and BL regions at -3 V versus pzc conditions.”**

Furthermore, we did not use fixed indices to tag the charge defects but computed those separately for each configuration used to sample the shown RDFs. This is now explicitly stated in the caption to Supplementary Figure 7 as follows: **As the charge defects migrate via structural (Grotthuss) diffusion, the specific oxygen**

sites were identified for all AIMD configurations to ensure that the defect RDFs are consistently computed in the presence of charge migration.

7. *Following the previous question, it would be better if the author could analyze RDF of Zundel and Eigen ions at different conditions separately to determine if the bias will affect solvation structures of these ions.*

Response: We fully agree and this has been done in response to previous question 6: “see **Supplementary Figure 7 for the comprehensive set of RDFs split into Eigen and Zundel states as well as IF and BL regions at -3 V versus pzc conditions.**”

8. *The authors stated that “Thus, we find that OH^- strongly interacts with electrified gold electrodes which is in line for instance with Raman spectroscopy finding that OH^- ions adsorb on Au surfaces.” According to the Raman spectroscopy (pH 9), the adsorbed OH is present at the potential range 0.40-1.0 V vs. Ag/AgCl. It is not clear what the pH condition the model corresponds to although the electrode potential has been reported. A more careful comparison is needed.*

Response: We thank the reviewer for highlighting the issue regarding the pH of the medium. To clarify, our simulation contains a single charge defect in the aqueous phase within the entire supercell setup of our systems. This is a standard approach used since many years in the context of atomistic AIMD simulations of excess charges in aqueous environments, see for instance Refs. [3, 24, 13], which does not provide access to properly determining the pH. The understanding is that this approach *forces* one excess charge defect, here OH^- , to be present in the aqueous phase such that its generic behavior can be investigated in the absence of any interference effects with other such charge defects in the sample (as required to address pH issues). In the present case, our simulation does for instance not provide information on what happens if more than one OH^- defect is in the IF region.

We have now addressed and clarified this point in the main manuscript: “**In practice, the pH of such solutions can influence the surface properties whereas our simulation set up is able to represent the intrinsic solvation structures of the individual H_3O^+ and OH^- charge defects in acidic and alkaline aqueous electrolyte interfaces but not the impact of concentration effects.**”

Minor points:

1. *Previous studies suggest that some density functionals (BLYP,PW91, HCTH) would yield qualitatively different solvation structures, diffusion constants. This manuscript should at least comment the possible effect of the functionals to the solvation structures. After all, there is no benchmark comparison to the experimental results during the study of the interfacial structures.*

Response: We are well aware of the point that different functionals might provide wrong structural diffusion scenarios of the charge self-defects in water[3, 23] as reviewed in Refs.[24, 13] We verified ourselves (without publishing it) that the RBPE-D3 functional we are using is very suitable for the purpose. Indeed, it has been shown recently by others that “the RPBE functional with D3 dispersion corrections provides a very accurate estimation of the diffusion coefficients of the excess proton and hydroxide ions in water, yielding a $D(\text{H}_3\text{O}^+)/D(\text{OH}^-)$ ratio that is equal to the experimental one.” (concluding statement in quotation marks cited from reference [15]).

We now comment in the methods part on this important technical point: “**It is well established that different density functionals can lead to incorrect structural diffusion scenarios for the charge self-defects in water^{3,23} as reviewed in Refs. [24, 13]. The RPBE functional with D3 dispersion corrections as used in this study has been shown recently to provide accurate estimates of the diffusion coefficients for excess protons and hydroxide ions in water, producing a $D(\text{H}_3\text{O}^+)/D(\text{OH}^-)$ ratio that matches the experimental value.¹⁵**”

2. I sometime got confused by the abbreviations used in this manuscript. For example, the IF for interface (also BL and IM). It would be better if the follow the tradition as those used in the study of metal/water interfaces.

Response: Metal/liquid interfaces can be divided into the inner and outer Helmholtz layers or are often classified as the interface region (first water layer) and the region beyond it. Alternatively, these regions can be defined as a function of distance, as shown in Revision Figure 3 using a representative example. We found it much easier to introduce well-defined shorthand notation instead of using “Region 1, Region 2, ... Region 6” or alike as introduced in other simulation studies of metal/water interfaces.

To enhance clarity and avoid any confusion, we now explain very explicitly the IF/IM/BL notation in the caption of Figure 1 with reference to markers in panels d and e: **“The interfacial (IF) and bulk-like (BL) regions are indicated by vertical dashed and dotted lines, respectively, while the intermediate (IM) region is in between as indicated by the respective labels in panels d and e.”** and, moreover, we refer in the main text to the definitions in this caption when our abbreviations IF, IM and BL are used for the first time.

Figure 1. Water density profiles and O/H number density profiles along the z -axis normal to the graphene sheets for systems XS to XL (see text), together with representative configuration snapshots at each confinement level; the graphene sheets are positioned exactly at the limits of the respectively shown z -ranges and all distributions have been symmetrized with respect to $z = 0$ Å. The vertical dashed lines in systems S, M, L, and XL at the minima of the water density profile visualize the partitioning between the different water layers, which are labeled as either interfacial (IF) or intermediate (IM) as explained in the text. Note that in system XL we distinguish between the outermost and the innermost IM water layers, indicating the latter by IM_i as opposed to plain IM for the former.

Revision Figure 3: Example of interfacial region partitioning in the literature. Figure taken from Ref.[18] (Reprinted with permission from Ref.[18] Copyright 2019 American Chemical Society).

3. For Figure 2 d-f, I think it will be better if they highlight the key structures and represent all other water molecules with smaller balls and sticks or just use sticks.

Response: We thank the reviewer for suggesting an alternative representation of the surrounding molecules. **We have now updated Figure 2 in the main text to use a stick representation as recommended by the referee.**

4. The presentation of the density profiles in Figure 1 d and e are also not easy to read. They can delete the density distribution of the metals and take the top of the Au surface atoms as reference plane.

Response: We liked this idea very much and revised all our respective figures in the manuscript and Supplementary Information. We now also reference the z -axis and thus the density profiles to the top gold layer (while shading the electrode rather than depicting the gold density distribution) following the suggestion of the referee.

5. It is also not clearly stated which surface is studied in the method section. And it takes me a while to realize that the author is studying Au(111) at line 568.

Response To provide this information much earlier, we have revised the introduction of the main text as follows: **“In this work, we implemented the latter method in the CP2K simulation package to investigate Au(111)/water interfaces in acidic and alkaline solutions ...”**.

Reviewer #3 (Remarks to the Author):

The manuscript “Distinct solvation patterns of OH^- versus H_3O^+ charge defects at electrified gold/water interfaces govern their properties” presents an ab initio molecular dynamics (AIMD) study of the interface between charged gold surfaces and liquid water. The specific focus of the manuscript is the interfacial solvation of hydroxide and hydronium species and how solvation is affected by applied electrode potentials. The simulations reveal that OH^- and H_3O^+ are solvated differently at the interface, with OH^- tending to reside in close contact with the gold surface and H_3O^+ tending to reside further away, approximately within the second solvent layer. These results and this contrast are interesting and may provide insight into the longstanding debate about the interfacial solvation properties of water ions. However, the explanation provided for these differences, i.e., that they reflect the response of ions to water’s underlying net atomic charge (NAC) profile, is not adequately supported by the data. Indeed, there appears to be a correlation between the positive regions of the NAC and the preferred location of OH^- , and (to a lesser extent) a correlation between the negative regions of the NAC and the preferred location of H_3O^+ . Whether this correlation is causal or incidental is not established. For this reason, I cannot recommend publication in Nature Communication. Below is a list of issues that the authors should consider upon revision.

Response: We appreciate the judgment that the reported results are interesting and may provide insight into the longstanding debate about the interfacial solvation properties of water ions. In the following, we will address each and every point of the referee to strengthen our conclusions and to improve our presentation.

1. Part of this study focuses on the influence of applied potential on the solvation properties of the water ions. The authors consider four conditions, pzc with OH^- , pzc with H_3O^+ , -3V with H_3O^+ , and $+1\text{V}$ with OH^- . I find it odd that the applied potential values are different for the hydroxide and hydronium case. Why did the authors not bias both at $+1\text{V}$ or $+3\text{V}$? Was -1V insufficient to draw H_3O^+ towards the interface?

Response: Let us explain our procedure first: At the start of the simulations, the bias potentials were gradually increased, starting from the lower bias potential (please see the Supplementary Figure 1). For the alkaline system, it was possible to sample the solvation structure of the OH^- ion in the interfacial water layer both at pzc and 1 V conditions. However, a larger negative bias potential was required to attract the H_3O^+ ion into the first water layer, so the referee is right when saying that “ -1V [was] insufficient to draw H_3O^+ towards the interface”. For this reason, $+1$ and -3V were used in the original manuscript for the two defect species; we note in passing that in the revised manuscript we have included additional simulations and analyses at -2V for the acidic case.

The important point is that these bias potentials generate close to the maximum voltages these two systems can experimentally support before electrochemical reactions set in according to Refs. [20, 4], thus we could not go more negative than -3V for hydronium and not more positive than $+1\text{V}$ for hydroxide to avoid electrochemistry going on in our simulation cell (which is not at all the focus of the present investigation). In order to clarify how the magnitude of the computational potentials can be compared to experimental voltages, we use surface charge densities as a bridge between experiment and simulations.

This is now explained in detail in the extended Supplementary Section 1 as follows: “**Based on the data in Supplementary Table 1, we can correlate the surface charge densities obtained from our calculations, based on the computational bias, with experimental values reported in the literature [20, 4]. Comparison to these experimental charge densities as a function of the electrode potential [20, 4] demonstrates that our computational voltages as defined above are in the regime that should not induce any electrochemical reaction in the acidic and alkaline cases. This is important since such reactions would strongly interfere with the intrinsic molecular properties of the H_3O^+ and OH^- charge defects close to the gold electrode on which we focus in this investigation.**”

2. I think it would be useful to see a comparison of the unbiased and biased OH^- and H_3O^+ distributions (e.g.,

in Figs. 1d and 1e). This would highlight an interesting contrast in the responsiveness of interfacial H_3O^+ vs OH^- to applied potential.

Response: We like this idea very much and now present a detail comparison of the unbiased and biased OH^- and H_3O^+ ion distributions. In response to this request, **we have now analyzed the corresponding density distributions and included the results in two new Supplementary Figures 9 and 10 with subsequent discussion.** The figures highlight the interesting contrast in the responsiveness of interfacial hydronium versus hydroxide by contrasting the distributions of these ions with and without bias potentials.

3. The definition of the CV for water ion position as presented in the SI is confusing to me. If I understand things correctly, then $n_i=2$ for all water molecules, so the summation in Eq. 2 would include many terms of $z_i \exp + -40$ and approximately 1 term with (hydronium $n_i=3$) $z_i \exp 60$ or (hydroxide $n_i = 1$) $z_i \exp -20$. I would expect the terms in the summation to look something like $z_i [1 - \exp \lambda(n_i - 2)]$.

Response: Indeed, we agree that this formula looks strange at first sight but it is fully correct as follows: Equation 2 in the Supplementary Information consists of many $z_i \exp(\lambda n_i)$ terms summed over the oxygen atoms of all water molecules in the sample. Among these terms, the ones corresponding to H_3O^+ or OH^- are weighted by a large factor and thus dominate, effectively determining the position of the two defects in space which is given by the collective variable s .

Because the summation in the numerator and denominator of Eq. 2 in the SI filters out smaller terms, the CV is not significantly affected by fluctuations in n_i . This results in a smooth function for identifying the positions of the respective defects. The smoothness and differentiability of Eq. 2 are crucial properties, ensuring that the CV successfully represents the positions of these charge defects in a continuous manner in time.

We have added more explanation in the Supplementary Information as follows: **“Regardless of the exact values of n_i , this specific functional form of the CV will effectively filter out smoothly everything but the position of the oxygen with the most (H_3O^+) or least (OH^-) amount of hydrogen in the required bond distance range, thus yielding analytically differentiable ion positions along AIMD trajectories.[17]”**

4. I find it very surprising that at least one of the hydrogens on H_3O^+ do not orient towards the electrode (like they do for water). Is there a precedent for this finding? Can the electronic structure calculation help to rationalize this finding? Maybe the shape of the MOs provide some insight?

Response: This question warrants detailed elaboration as follows which, in short, supports our findings. A recent experimental study using Vibrational Sum Frequency Generation (VSFG) spectroscopy successfully identified the H_3O^+ core in an Eigen-like configuration at negatively charged interfaces [25] This study concluded that the C_3 axis of H_3O^+ is aligned along the surface normal, as shown in Figure 3 of the study and reproduced in the top panel of Revision Figure 4. If one of the hydrogens H_3O^+ oriented itself toward the electrode, as suggested by the referee, the C_3 axis of H_3O^+ would be accordingly tilted and no longer be normal to the surface.

Secondly, another study [22] visualized the H_3O^+ configuration drawing on AFM measurements and DFT calculations, where H_3O^+ is located slightly above the water layer. In this study, the Au surface was also negatively charged due to the presence of excess H^+ in the monolayer of water on the Au surface. Consequently, the hydrogen atoms of water molecules were oriented toward the Au surface, as depicted in Figure 1f of the study.[22] Yet, it was found that the hydrogen atoms of the H_3O^+ charge defect do *not point* toward the Au surface but instead participate in hydrogen bonding with surrounding water molecules exactly as we find it in our AIMD simulations if H_3O^+ is slightly further away from the surface.

To further elucidate the orientation of H_3O^+ in response to the question of the referee, **we conducted additional**

analysis of the probability distribution shown in the new Supplementary Figure 6 (reproduced as bottom panel in Revision Figure 4), which characterizes the orientation of H_3O^+ within the interfacial (IF) region normal to the surface along z .

To highlight the current understanding concerning the orientation of H_3O^+ close to the Au surface which support our fully atomistic findings that greatly transcend the existing experimental insights, we have incorporated the following discussion into the Supplementary Information following the new figure:

“In case that the oxygen site of the H_3O^+ defect is rather close to the Au surface, around $z \approx 3 \text{ \AA}$ according to Supplementary Figure 6, its oxygen atom preferentially faces the Au surface, with its C_3 axis aligned to the surface normal, while its three hydrogen atoms form hydrogen bonds with neighboring water molecules, thus not pointing toward the electrode surface as illustrated in the representative top-left snapshot in Supplementary Figure 6. This finding is in agreement with Vibrational Sum Frequency Generation (VSFG) spectroscopy concluding that the C_3 axis of H_3O^+ is aligned along the surface normal.²⁵ Alternatively, if one of the three H_3O^+ hydrogens oriented itself toward the electrode, the C_3 axis of H_3O^+ would be accordingly tilted and no longer perpendicular to the surface.

In contrast to this scenario, when H_3O^+ is located further away from the topmost gold layer (where $z = 0 \text{ \AA}$), the most probably orientation of the C_3 axis changes toward being perpendicular to the surface normal, and thus parallel to the gold surface, see Supplementary Figure 6. As seen from the real-space configuration depicted in the top-right snapshot, H_3O^+ acts now as a bridge between water molecules at direct proximity to the Au surface and those that are further away. Note that the interface (IF) water layer extends up to roughly 4.3 \AA . Interestingly, this orientation closely resembles the H_3O^+ configuration investigated by AFM experiments and DFT calculations,²² where H_3O^+ is located slightly above a monolayer of water on the Au surface.”

Fig. 3 Polarization dependence and orientational analysis of the hydrated Eigen proton. **a** VSF spectra collected in the polarization combinations SPS, SSP, and PPP of a dAA monolayer on a 1 μ M NaCl solution in H₂O (surface pressure 20 mN m⁻¹). **b** Theoretical curves showing the expected variations of the SF intensity of the antisymmetric stretch of H₃O⁺ (C_{3v} symmetry) as a function of tilt angle of the C₃ axis from the surface normal for the three polarization combinations, assuming a delta distribution (solid lines), and Gaussian distribution with standard deviation of 15° (dashed lines). See Supplementary Note 2 for details of the model. The sketch depicts the orientation of the hydronium core with the arrow along the C₃ axis. Experimental observations are consistent with the C₃ axis aligned closely to the surface normal. In the sketch, the oxygen and hydrogen atoms of the core Eigen proton are depicted as blue and red spheres, respectively. In the flanking water molecules, oxygen atoms are in light blue, while hydrogens are illustrated as white spheres.

Revision Figure 4: Figures taken from the VSGF experiment [25] that provides a scenario for H₃O⁺ at the gold-water interface that is consistent with our findings from the main text (and analyzed in more detail in the bottom panel). Bottom: Probability distribution functions of the cosine of the angle θ between the vector sum of the O-H bonds in H₃O⁺ and the surface normal as a function of z . **This additional analysis is added as Supplementary Figure 6 to the revised Supplementary Information together with extensive discussion.**

5. The authors state on page 5 that “Clearly, the positive potential bias drives the OH^- defect into the first water layer,...”. Based on the results at pzc, this statement is misleading. The OH^- appears to be driven to the first water layer even without a positive potential bias.

Response: We feel that our discussion, resulting in that statement, did not provide the full picture. In order to improve our presentation, we now elaborated on the migration scenario of OH^- and its difference compared to H_3O^+ as follows: “Clearly, the positive potential bias drives the OH^- defect—initially located deep in the BL regime, approximately 10 Å from the topmost Au layer at time zero in the migration pathway (Fig. 1c)—into the IF region, which extends up to about $z \approx 4.4$ Å (as indicated by the water density profile in Fig. 1e). The defect crosses this distance at $t \approx 27$ ps (Fig. 1c) before finally localizing just $z \approx 2$ Å from the surface. We find that the interfacial water structure itself as quantified by the O and H density profiles in Fig. 1e is not impacted by the 1 V bias compared to the pzc reference (see Supplementary Fig. 10 for refined analysis). This OH^- migration scenario is distinctly different from H_3O^+ where, subject of a bias of -3 V, this defect barely moves into the IF region. Rather it ends up in the IM regime or at best in between the first and second solvation layer as observed for $t > 20$ ps in the representative pathway of Fig. 1c.”

6. If the charge Δq oscillations for water drive the interfacial solvation properties of these species, then I would expect the density profiles in Fig. 1d and 1e for OH^- and H_3O^+ to roughly mirror the oscillations of Δq . For instance, if region of positive NAC is responsible for the solvation position of OH^- , then why isn't there a secondary OH^- peak around $z=12$, where there is another positive NAC region?

Response: As a basis for this discussion we have carried out additional analyses of the density profiles of oxygen atoms in water molecules together with the charge defects that we report in the new Supplementary Figures 9 and 10 for H_3O^+ and OH^- , respectively.

To highlight the propensities of H_3O^+ and OH^- , we have introduced the following discussion in relation to the new Supplementary Figures 9 and 10:

“Supplementary Figures 9 and 10 present the density profiles of oxygen atoms in water molecules and in the two charge defects normal to the surface along z along with the corresponding excess charge profiles. The first figure indicates that H_3O^+ is able to approach the negative region of the excess charge around roughly 3 Å whereas its density profile in the IF region peaks at slightly larger distances where the excess charge starts to become positive. The free energy barrier for proton transfer near the surface suggests that the presence of the Au surface does not hinder proton transfer compared to the bulk phase (see main text, in particular Figure 3d: compare IF to BL data). This behavior can be attributed to the flat solvation structure of H_3O^+ , which can perfectly integrate itself into the existing two-dimensional hydrogen bond network near the surface via strong hydrogen bonding (given the essentially perpendicular arrangement of the C_3 axis of the hydronium ion). The resulting barrierless proton transfer process enables H_3O^+ to migrate in the region which corresponds to the first negative region of the NAC near the surface (around roughly 3 Å). However, at slightly larger distances, there is a change in orientation of H_3O^+ toward a more parallel arrangement of its C_3 axis w.r.t. the surface (see Supplementary Figure 6) where the defect forms a bridge from the first to the second layer, being the region where the excess charge switches from negative to positive.

The OH^- ion is primarily localized around ≈ 2 Å which is deep in the interfacial (IF) region where the excess charge is positive, see Supplementary Figure 10. We note that the excess charge is also positive around roughly 4 Å where no OH^- density is found, however there is simply not enough water at this distance that delimits the IF region that could host the defect. Once the water density is again sufficiently high beyond that first minimum, the propensity to find OH^-

again correlates with the excess charge profile, for instance an increased population close to 7 Å is observed which is particularly pronounced at finite bias.”

In addition to this detailed analysis, we now extended the abstract in this respect by pointing out that the charge defects must also be able to integrate themselves in the water network given their distinct solvation patterns by adding: “... **in concert with the distinct solvation patterns of these charge defects.**” and specifically referring to hydrogen bonding: “Due to the combined effects of this net atomic charge, **hydrogen bonding**, and ...”. Similarly, we now point out the importance of solvation also in the revised Discussion section as marked therein, see also our feedback to the next much related question.

7. In the discussion section, I think claiming that the oscillatory feature of the NAC “repels H_3O^+ ions from the first layer as we demonstrate” is much too strong of a statement. In fact, I’m sure much of the difference between the H_3O^+ and OH^- in these systems has to do with the solvation (or desolvation) free energetics.

Response: In addition to the changes in response to the closely related previous question, we now clarify the situation concerning additional factors beyond the NAC profiles, in particular **we now avoid the too strong statement when using the word “repel” and make clear that also solvation, meaning in particular the specific H-bonding interactions of the respective charge defects, also contribute to the full picture.**

In response to this question, we modified the Discussion section accordingly: “... **can be attributed to not only the positive hydrogen binding energy, but also to the oscillatory feature of the NAC at the interface that disfavors H_3O^+ ions to localize in the first layer. Importantly, we note that it has been reported that H_3O^+ can integrate well into bilayer water structures,¹⁴ indicating that the solvation structure alone cannot fully explain the preferential location of this ion, yet the defect must be able form HBs to be hosted. Additionally, a previous study employing STM and DFT²¹ found that H_3O^+ resides predominantly in the second layer. These findings collectively suggest that the oscillatory NAC profile plays a pivotal role at liquid/metal interfaces together with proper solvation of the specific charge defects within the local HB network of interfacial water.**” and further up in the Discussion: “... **the oscillatory excess charge within water near the Au surface in concert with hosting the new solvation structure of this charge defect that requires specific H-bonding.**”. Recall that we also extended the abstract to stress already therein that proper solvation of the specific charge defects is also required, see above.

References

- [1] Andrea Auer et al. “The potential of zero charge and the electrochemical interface structure of Cu (111) in alkaline solutions”. In: *J. Phys. Chem. C* 125.9 (2021), pp. 5020–5028.
- [2] Assil Bouzid and Alfredo Pasquarello. “Atomic-scale simulation of electrochemical processes at electrode/water interfaces under referenced bias potential”. In: *J. Phys. Chem. Lett.* 9.8 (2018), pp. 1880–1884.
- [3] Amalendu Chandra, Mark E Tuckerman, and Dominik Marx. “Connecting Solvation Shell Structure to Proton Transport Kinetics in Hydrogen–Bonded Networks via Population Correlation Functions”. In: *Rev. Lett.* 99.14 (2007), p. 145901. DOI: 10.1103/PhysRevLett.99.145901.
- [4] Aicheng Chen and Jacek Lipkowski. “Electrochemical and spectroscopic studies of hydroxide adsorption at the Au (111) electrode”. In: *J. Phys. Chem. B* 103.4 (1999), pp. 682–691.
- [5] Benoît Grosjean, Marie-Laure Bocquet, and Rodolphe Vuilleumier. “Versatile electrification of two-dimensional nanomaterials in water”. In: *Nat. Commun.* 10.1 (2019), p. 1656.
- [6] Jongwon Kim and Andrew A Gewirth. “Mechanism of oxygen electroreduction on gold surfaces in basic media”. In: *J. Phys. Chem. B* 110.6 (2006), pp. 2565–2571.
- [7] Rasmus Kronberg and Kari Laasonen. “Dynamics and surface propensity of H⁺ and OH[−] within rigid interfacial water: Implications for electrocatalysis”. In: *J. Phys. Chem. Letters* 12.41 (2021), pp. 10128–10134.
- [8] Jinggang Lan, Vladimir V Rybkin, and Marcella Iannuzzi. “Ionization of water as an effect of quantum delocalization at aqueous electrode interfaces”. In: *J. Phys. Chem. Lett.* 11.9 (2020), pp. 3724–3730.
- [9] Chao-Yu Li et al. “In situ monitoring of electrooxidation processes at gold single crystal surfaces using shell-isolated nanoparticle-enhanced Raman spectroscopy”. In: *J. Am. Chem. Soc.* 137.24 (2015), pp. 7648–7651.
- [10] Peng Li et al. “Hydrogen bond network connectivity in the electric double layer dominates the kinetic pH effect in hydrogen electrocatalysis on Pt”. In: *Nat. Catal.* 5.10 (2022), pp. 900–911.
- [11] Xiao Li and Andrew A Gewirth. “Peroxide electroreduction on bi-modified au surfaces: vibrational spectroscopy and density functional calculations”. In: *J. Am. Chem. Soc.* 125.23 (2003), pp. 7086–7099.
- [12] Pietro P Lopes et al. “Double layer effects in electrocatalysis: The oxygen reduction reaction and ethanol oxidation reaction on Au (1 1 1), Pt (1 1 1) and Ir (1 1 1) in alkaline media containing Na and Li cations”. In: *Catal. Today* 262 (2016), pp. 41–47.
- [13] Dominik Marx, Amalendu Chandra, and Mark E Tuckerman. “Aqueous basic solutions: hydroxide solvation, structural diffusion, and comparison to the hydrated proton”. In: *Chem. Rev.* 110.4 (2010), pp. 2174–2216.
- [14] Daniel Munoz-Santiburcio and Dominik Marx. “On the complex structural diffusion of proton holes in nanoconfined alkaline solutions within slit pores”. In: *Nat. Commun.* 7.1 (2016), p. 12625.
- [15] Daniel Muñoz-Santiburcio. “Accurate diffusion coefficients of the excess proton and hydroxide in water via extensive ab initio simulations with different schemes”. In: *J. Chem. Phys.* 157.2 (2022).
- [16] Minoru Otani et al. “Electrode dynamics from first principles”. In: *J. Phys. Soc. Jpn.* 77.2 (2008), p. 024802.
- [17] Jung Mee Park et al. “Dissociation mechanism of acetic acid in water”. In: *J. Am. Chem. Soc.* 128.35 (2006), pp. 11318–11319.
- [18] Sergi Ruiz-Barragan, Daniel Munoz-Santiburcio, and Dominik Marx. “Nanoconfined water within graphene slit pores adopts distinct confinement-dependent regimes”. In: *J. Phys. Chem. Lett.* 10.3 (2018), pp. 329–334.
- [19] Sung Sakong and Axel Groß. “The electric double layer at metal-water interfaces revisited based on a charge polarization scheme”. In: *J. Chem. Phys.* 149.8 (2018), p. 084705.

-
- [20] Z Shi et al. “Investigations of SO_4^{2-} adsorption at the Au (111) electrode by chronocoulometry and radiochemistry”. In: *J. Electroanal. Chem.* 366.1-2 (1994), pp. 317–326.
- [21] Felice C Simeone et al. “Tunneling behavior of electrified interfaces”. In: *Surf. Sci.* 602.7 (2008), pp. 1401–1407.
- [22] Ye Tian et al. “Visualizing Eigen/Zundel cations and their interconversion in monolayer water on metal surfaces”. In: *Science* 377.6603 (2022), pp. 315–319.
- [23] Mark E Tuckerman, Amalendu Chandra, and Dominik Marx. “A statistical mechanical theory of proton transport kinetics in hydrogen-bonded networks based on population correlation functions with applications to acids and bases”. In: *J. Chem. Phys.* 133.12 (2010).
- [24] Mark E Tuckerman, Amalendu Chandra, and Dominik Marx. “Structure and dynamics of $\text{OH}^-(\text{aq})$ ”. In: *Accounts of chemical research* 39.2 (2006), pp. 151–158. DOI: 10.1021/ar040207n.
- [25] Eric Tyrode, Sanghamitra Sengupta, and Adrien Stoeber. “Identifying Eigen-like hydrated protons at negatively charged interfaces”. In: *Nat. Commun.* 11.1 (2020), pp. 1–7.

REVIEWER COMMENTS

The changes to the manuscript (MS) and supplementary information (SI) that result from the present reviewing process are **marked in blue therein** while we keep all changes from the first review process therein **marked using red color** as originally. The present changes to the MS and SI are referred to in our point-by-point replies below **using boldface fonts**.

Reviewer #1 (Remarks to the Author):

While the authors agree with my comments on the limitations of the constant potential method, they tend to omit these very important aspects from the main manuscript. This work is a first application of Neugebauer’s method after implementation in cp2k code. Cp2k is a widely used code and I am afraid if the authors do not include a clear word of caution about the limitations of the potentiostat method and specifically that it is not a method suitable for modelling electrochemical cells, this will induce unexperimented future users into error. Not everyone is aware of these limitations and most of users will tend to apply it blindly. Therefore, I invite the authors to add a clear and honest statement about the limitations of this technique in a future revision.

Response:

Indeed, we did not intend in the present work to model an electrochemical cell in accord with the judgement of the Reviewer.

In response to the Reviewer’s request, we now provide in the SI a detailed discussion of the technique, followed by clear and honest statements about its limitations. The latter are unfolded in detail in the revised SI Section 1 by adding a full page of discussion and explanation of the problems and limitations to Section 1 (blue text).

In the MS, we added a concise version of that discussion on about half a page and spell out the limitations and caveats of the methodology also in the main text, starting with the blue sentence “Here, we employ this technique to carry out AIMD simulations ...”, while we refer to the SI for a comprehensive discussion; in the preceding paragraph in the MS starting with “Likewise, comprehensive ...” we now refer to more computational methods than previously to provide a broader embedding.

This method cannot fundamentally simulate correctly a bias-potential and its corresponding charge density at the electrode surface. This means that the bias potential reported by the authors does not correspond to the reported charge density (cannot predict capacitance!). Given this fundamental limitation, the authors suggest to simply use the surface charge density to define their systems. If this choice is kept, the paper should then be reshaped to reflect that these are not constant-potential simulations and not even a proper way to model electrochemical cells. As such, this method can only help to access reaction mechanisms at various surface charge densities without quantitatively accessing the electrochemical properties of the interface in a way comparable to experiments. The conclusions must then be tuned accordingly.

Response:

We are aware of the points raised and thus fully agree on what the Reviewer concludes and suggests us to do. Importantly, “we kept our choice” of using the surface charge density – as offered by the Reviewer – for reasons that we now explain more clearly, and thus “reshaped the paper” throughout while closely following the requests of the Reviewer.

In the SI, after having discussed the problems, caveats and limitations of the method (and also in the MS), see our response to the first point, we additionally address in the SI in much detail (blue text) the issue of using the surface charge density to define our systems, which is supported by

additional analysis presented in the new SI Figure 2, while being *“unable to predict capacitance”*. In the same vein, we now spell out that we *“cannot quantitatively access the electrochemical properties of the interface in a way comparable to experiments”*, and we also make clear that *“this method can only help to access reaction mechanisms at various surface charge densities”*.

In the MS, we change our presentation throughout the entire text (see lots of blue text in all sections and figure captions) in the sense that we now refer to different surface charge density conditions that we generate by applying the corresponding computational bias potentials. We also make clear that we cannot model full electrochemical cells, which would allow for quantitative comparison to experiment, and thus use the words “qualitative” and “trends” where we refer to comparisons to experiment (see also blue highlightings). Still, in full agreement with the assessment of the Reviewer, we can qualitatively access the impact of various surface charge densities on reaction mechanism, which are in our case the charge transfer processes of H_3O^+ versus OH^- that underlie their Grotthuss-like migration at gold/water interfaces.

Reviewer #2 (Remarks to the Author):

The authors have made efforts to address the comments raised. I am fine with most of them, ...

Response:

We appreciate very much that the Reviewer is satisfied with how we addressed the vast majority of the previous comments.

... but still not convinced of the argument on the unphysically too negative electrode potential. Previous AIMD work in literature can well reproduce the capacitance, i.e. relation between surface charge density and electrode potential. Why is there such a large discrepancy that leads to the very negative potential -3 V, estimated from the surface charge density. Also, it seems that at -3V the OH angle and RDF of the ions are not affected much compared to the PZC condition. I am very concerned on how the electrode potential is determined and how accurate the interfacial structures are.

Response:

The only concern left is related to the nominal values of the computational electrode potentials as reported by us and their relation to the computed surface charge densities, providing the basis to judge the impact of non-zero surface charge densities on properties such as interfacial structures.

We are aware of outstanding previous computational work that nicely reproduces experimental surface charge densities and the dependence on the applied bias potential (we now amended the references to previous work in the introduction before introducing our approach). In our present setup, we deliberately exclude any electrolyte ions from our acidic and alkaline aqueous solutions, which exclusively contain the following charge defects, H_3O^+ or OH^- , and their respective counterions (located close to the computational neon counter electrode). This particular approach ensures that the water structure at the metal electrode, in particular its interfacial hydrogen bond network, does not get perturbed by the presence of such electrolyte ions, since ions at aqueous surfaces are known to strongly alter the interfacial water structure due to the need to establish their own hydration shells. Here, they would strongly interfere with the solvation shell structure of the H_3O^+ and OH^- charge defects, which we would like to investigate and understand in the absence of such (concentration-dependent) perturbations.

As a consequence of our choice, no electric double layer can form in our supercell setup, which would however be fundamentally necessary to compute the proper dependence of the surface charge density on the applied electrode potential as measured in typical electrochemical experiments via the double-layer capacitance. This implies that the bias potential should be viewed in our approach as a computational parameter which allows us to control, in a continuous manner, the surface charge density as we now demonstrate in the new SI Figure 2 (see also the enhanced discussion and justification of our current computational setup in the revised SI Section 1, blue text). The new Figure 2 in the SI shows that our computational bias of -3 V translates into a surface charge density of *roughly* $\sigma \approx -13(1) \mu\text{C cm}^{-2}$ (see also the compiled computational data in SI Table 1). Although we cannot rigorously compare our computational bias potentials to experimental voltages, we can *qualitatively* compare the computed surface charge densities to corresponding experiments at low concentrations [1], see Revision Figure 1. Based on this figure, showing measured surface charge density versus electrode potential curves for acidic 0.1 M HClO_4 solutions containing varying concentrations of electrolyte ions at the Au(111) electrode, one can see that our computed surface charge densities are *qualitatively* in the proper voltage regime before electrochemical reactions set in (which we clearly want to avoid given the purpose of the present study).

We conclude that our most negative computational bias potential in case of the simulated acidic HClO_4 solution corresponding to a surface charge density of *about* $-13(1) \mu\text{C cm}^{-2}$ is *qualitatively* consistent with experimental low concentration data for 0.1 M HClO_4 electrolyte solutions which yield an experimental surface charge density of $\sigma_{\text{M}} \approx -(16 - 17) \mu\text{C cm}^{-2}$ at the most negative electrode potential according to Revision Figure 1.

In response to this question, we now added to the SI a comprehensive discussion (blue text) of

Fig. 2. Charge density–electrode potential curves at the Au(111) electrode for the 0.1 M HClO₄ solution (○) with the following additions of K₂SO₄: ● 5 × 10⁻⁶ M K₂SO₄; □ 2.5 × 10⁻⁵ M K₂SO₄; ■ 5 × 10⁻⁵ M K₂SO₄; △ 10⁻⁴ M K₂SO₄; ▲ 5 × 10⁻⁴ M K₂SO₄; ◇ 2.5 × 10⁻³ M K₂SO₄; ◆ 5 × 10⁻³ M K₂SO₄. For clarity only the data for a few concentrations are shown in the figure.

Revision Figure 1: Experimental surface charge density versus electrode potential curves for acidic 0.1 M HClO₄ aqueous electrolyte solutions with low concentrations of K₂SO₄ salt at the Au(111) electrode [1] (Reprinted from Ref.[1], Copyright (1994), with permission from Elsevier).

this important aspect at the end of the paragraph in SI Section 1 starting with “Based on the data ...” right after having introduced and discussed SI Table 1.

Also, it seems that at -3V the OH angle and RDF of the ions are not affected much compared to the PZC condition. I am very concerned on how the electrode potential is determined and how accurate the interfacial structures are.

Response:

We would like to remark that the water structure we obtain close to the charged gold electrode does change significantly at the corresponding surface charge density as evidenced, in the first place, at the level of both the oxygen and hydrogen density profiles as depicted in Figure 1(d) in the MS: The double-peaked structure of the interfacial hydrogens emerges in the charged case with reference to their unimodal peak in the absence of bias (pzc data). In case of the oxygens, their first solvation layer peak in the zero bias case (pzc) is broad and skewed subject to a pronounced shoulder toward the gold electrode, whereas that peak gets both sharpened and shifted away from the negatively charged surface. Overall, the first solvation layer of water molecules features distinct structural responses upon negatively charging the gold electrode as explained in the MS. In case of the H₃O⁺ orientations, we wish to point out the suppressed probability of θ angles at less than about 30° in the Eigen state and the clear enhancement of $\theta \approx 90^\circ$ orientations for Zundel-like complexes compared to their pzc references based on the data in Figure 2(a) and (b) in the MS.

In response to this part of the Reviewer’s question as to structural changes upon charging, we amended the discussions in the revised MS when (i) analyzing the water structure based on Fig-

ure 1(d) by pointing out the distinct structural response (in the paragraph starting with “How does the H_3O^+ charge defect integrate ...”) and (ii) discussing the changes of H_3O^+ orientations based on Figure 2(a) and (b) (in the paragraph starting with “What is the orientation ...”).

A typo: At line 171, “as a has been realized using a continuous”

Response:

We thank the reviewer for pointing this out to us. The error has now been corrected.

Reviewer #3 (Remarks to the Author):

The revisions and responses have addressed some of my previous concerns. The results highlight some interesting differences in the interfacial solvation of H_3O^+ and OH^- .

Response:

We thank the Reviewer for pointing out that our revision and responses have addressed previous concerns and that our results highlight interesting differences in the interfacial solvation of H_3O^+ and OH^- .

However, I think the results lack the scientific novelty to warrant publication in Nature Communications. The primary conclusions, e.g., that OH^- coordinates to the gold surface, have an experimental precedent. Highlighting specific details of the solvation structure, e.g., orientational angles, and their subtle dependences on applied potential certainly merit publication, but these results do little to transform the way we think about the electrochemical processes that these species are involved in. Thus, I recommend the authors submit this manuscript to a more specialized journal.

Response:

In order to address the remaining criticism, we now explicitly outline the scientific novelty of our findings:

- We demonstrate and explain why the hydrated hydronium ion, H_3O^+ , avoids the first solvation layer at both uncharged and negatively charged gold electrodes while OH^- has a strong propensity to populate the interfacial water layer even in the absence of charging. Our mechanistic explanation goes down at the electronic structure level. Thus, we do not merely “highlight some interesting differences in the interfacial solvation of H_3O^+ and OH^- ” as suggested by the Reviewer, but we first of all demonstrate very distinct differences between the hydronium and hydroxide ions at electrified gold/water interfaces and, more importantly, also explain them at the most fundamental level.
- We find that the properties of hydronium at this metal/water interface remain unaltered with respect to those in the bulk limit irrespective of charging the surface, while those of hydroxide are distinctly different at the gold/water interface from what is known from bulk water. In case of OH^- , we discover that the celebrated hypercoordinated “resting state” as well as the “active state” for charge transfer as known from the bulk environment are both significantly altered by involving Au atoms, instead of water molecules, to stabilize the hydrated proton hole at the gold/water interface along charge transfer processes. And again, we trace this novel finding back to changes in the electronic structure of OH^- interacting with the metal surface versus being embedded in the bulk water environment. This has not been discovered before and evidently also not explained. Clearly, this goes much beyond just discussing “subtle dependences on applied potential” as the Reviewer writes.
- We find that the charge transfer properties of the OH^- charge defect are significantly altered while those of H_3O^+ remain unchanged compared to their bulk water references. Firstly, a novel Grotthuss-like mechanism is found that critically involves gold atoms from the electrode – again never published before. Secondly, this specific mechanism leads to a significant increase of the charge transfer barrier. This slows down very much the charge transfer rate involving OH^- within interfacial water with reference to bulk water (in stark contrast to H_3O^+) by close to an order of magnitude according to transition state theory. Also this finding is novel for charge transfer at alkaline versus acidic gold/water interfaces.
- All these findings are not merely “Highlighting specific details of the solvation structure”, but impact significantly on our fundamental understanding of the outstanding differences of acidic versus alkaline aqueous solutions in contact with both uncharged and charged metal electrodes.

We are unaware that all these important findings – being certainly not “*specific details*” and “*subtle dependencies*” as the Reviewer writes – have been communicated in the extant literature, both experimental and computational.

In the revised MS, we now more clearly and comprehensively spell out the novelty of our findings in the abstract (blue sentences at the end), introduction (blue sentences at the end of the last paragraph before “*Results*”), results section (blue discussion starting with “*Overall, we find that the free energy barrier to charge transfer ...*”) and conclusions (in the second last paragraph starting with “*In conclusion, we ...*”) in addition to what we already described as our findings in the previous version.

References

- [1] Z Shi et al. "Investigations of SO_4^{2-} adsorption at the Au (111) electrode by chronocoulometry and radiochemistry". In: *J. Electroanal. Chem.* 366.1-2 (1994), pp. 317–326.

REVIEWER COMMENTS

The changes to the manuscript (MS) and supplementary information (SI) that result from the present reviewing process are **marked in orange therein** while we keep all changes from the first and second review process therein **marked using red and blue** colors as originally. The present changes to the MS and SI are referred to in our point-by-point replies below **using boldface fonts**.

Reviewer #1 (Remarks to the Author):

The authors addressed all my concerns and provided a clear and detailed explanation of the limitations of their methods. The extensive computational efforts in delivering high-quality first-principles modelling and the novel mechanistic aspects discovered by the authors convinced me that this work is worth publication in Nature Communications journal.

Response:

We sincerely appreciate the Reviewer's positive assessment and are pleased that **our discussion of the methodological limitations was found to be clear and detailed in the revised version**.

Reviewer #2 (Remarks to the Author):

The authors have made further revisions to their original draft and most of my concerns have been addressed in this version. I would recommend its publication in Nature Communication provided that the following minor revisions are made.

Response:

We are grateful to the Reviewer for their overall positive feedback and are pleased that the **majority of concerns have been resolved in the further revisions of the original draft**. We are happy to now address the remaining **two minor revisions**:

1. The author should explicitly outline the potential issues associated with electrode potential calculations in this paper. This will prevent readers from mistakenly assuming that this method can be utilized for electrochemical studies without additional critical evaluation.

Response:

We thank the Reviewer for raising this point regarding the electrode potential calculations. **To clarify these potential issues, we have amended the manuscript as follows:**

Because of the lack of screening counterions in the electric double layer, the computational voltage does not represent the local voltage drop near the metal electrode. However, these features would be essential for modeling realistic electrochemical cells, where a well-defined relationship between electrode potential and surface charge density is required. Hence, the underlying limitations must be taken into account when using a similar computational setup in future electrochemical studies. In contrast to such applications, our aim here is to set up an electrified interface between an explicit gold electrode model in contact with pure water, enabling us to investigate—depending on surface charge density—the intrinsic structural and dynamical properties of H_3O^+ and OH^- charge defects. ...

as well as in the Supporting Information:

In addition, the absence of a double layer at the interface results in incomplete screening of the electrostatic field from the Au surface. The computational voltage, which is calculated using the dipole correction within

the present approach, does not correspond to the voltage drop across the double layer as measured in typical electrochemical experiments. Nevertheless, this *computational* voltage can still serve as a measure of the applied bias in the system: As we demonstrate below in detail, the surface charge density can be controlled systematically and continuously by changing the *computational* voltage although the quantitative relation of applied voltage and surface charge density cannot be accessed, leaving comparisons to experiments at the qualitative level; see below for a detailed discussion of the caveats of the present approach as used here and their implications. Despite these clear limitations, the current AIMD-based setup does allow us to ...

We hope that these very explicit additions *will prevent readers from mistakenly assuming that this method can be utilized for electrochemical studies without additional critical evaluation.*

2. *In Figure 3a and 3b, I recommend that the authors include RDF data for H₂O-H₂O as reference. This addition will allow readers to observe the relative peak position shifts surrounding H₃O⁺/OH⁻.*

Response:

We like the suggestion of the Reviewer and **now include the corresponding RDF data for H₂O–H₂O as reference in the revised Figures 3a and 3b and amended the caption accordingly** that *will allow readers to observe the relative peak position shifts surrounding H₃O⁺/OH⁻.*